# Control, Transport and Sampling:
# The Benefit of a Reference Process

## Abstract

We aim to establish connections between diffusion-based sampling, optimal transport, and optimal (stochastic) control through their shared links to the Schrödinger bridge problem. Throughout, we highlight the importance of having a reference measure on the path space for the design of a valid objective function that can be used to transport $\nu$ to $\mu$, consequently sample from the target $\mu$, via (optimally) controlled dynamics.

## 1 Introduction

Traditionally, the task of sampling from un-normalized density is largely delegated to MCMC methods. However, modern machine learning developments in optimal transport and generative modeling have greatly expanded the toolbox we have available for performing such tasks, which can further benefit from powerful advancements in neural networks. It holds promise for sampling from high-dimensional and multimodal distributions compared to MCMC-based alternatives. In this work, we propose control objectives that are amenable to tractable approximation (without access to data from the target, as classically considered in the MCMC literature) for sampling purpose. The central object for this synthesis will be forward-backward SDEs via time-reversals.

## 2 General Framework

In this section we put several recently proposed (and diffusion-inspired) methods in context, rendering them as special instantiations of a more unifying *path-wise* picture. We loosely follow the framework put forth in Vargas & Nüsken (2023) for some parts.

### 2.1 Goal and Setup

We are interested in sampling from $p_{\text{target}}(x) = \mu(x)/Z$ by minimizing certain tractable loss, assuming sampling from $\nu(z)$ is easy, but unlike in generative modeling, even though analytical expression for $\mu$ is readily available, we do not have access to data from it that can be used to learn the score function. In this sense, in terms of "transport mapping", the two sides are crucially not symmetric. More specifically, the following represents a coupling between $p_{\text{prior}}(z) = \nu(z)$ and $p_{\text{target}}(x)$, our desired target distribution, where the transitions are parametrized by $\gamma$ and $\theta$:

$$\pi(x,z) := q^{\gamma}(z|x)p_{\text{target}}(x) = p^{\theta}(x|z)p_{\text{prior}}(z).$$

Minimizing objective of the type ($D_{KL}$ below denotes KL divergence)

$$\mathcal{L}(\gamma,\theta) := D_{KL}(q^{\gamma}(z|x)p_{\text{target}}(x)\|p^{\theta}(x|z)p_{\text{prior}}(z))$$

to 0 will ensure the marginals are satisfied, although the minimizers of the problem will not be unique. More generally, one can make the transitions more hierarchical (yet still Markovian), and for this we introduce

forward-backward SDEs. Pictorially, given a base drift $f$, we have the "sampling process" (the two processes are time reversals of each other):

$$\nu(z) \underset{\mathbb{P}^{\mu,f+\sigma v}}{\overset{\mathbb{P}^{\nu,f+\sigma u}}{\rightleftharpoons}} \mu(x) \tag{1}$$

for tunable controls $u, v$ and terminal marginals $\nu, \mu$. Written as SDEs, they become (the drifts $u, v$ here are not independent)

$$dX_t = [f_t(X_t) + \sigma u_t(X_t)]dt + \sigma \overrightarrow{dW_t}, \ X_0 \sim \nu \Rightarrow (X_t)_{t \in [0,T]} \sim \overrightarrow{\mathbb{P}}^{\nu,f+\sigma u}, \tag{2}$$

$$dX_t = [f_t(X_t) + \sigma v_t(X_t)]dt + \sigma \overleftarrow{dW_t}, \ X_T \sim \mu \Rightarrow (X_t)_{t \in [0,T]} \sim \overleftarrow{\mathbb{P}}^{\mu,f+\sigma v}. \tag{3}$$

Operationally, (2)-(3) denote (picking $f = 0$, and $\{z_t\}$ a sequence of i.i.d standard Gaussians for illustration)

$$X_t = X_0 + \int_0^t \sigma u_s(X_s)ds + \int_0^t \sigma \overrightarrow{dW_s} \Rightarrow X_{t+h} \approx X_t + h\sigma u_t(X_t) + \sqrt{h}\sigma z_t, \quad X_0 \sim \nu \tag{4}$$

$$X_t = X_T - \int_t^T \sigma v_s(X_s)ds - \int_t^T \sigma \overleftarrow{dW_s} \Rightarrow X_{t-h} \approx X_t + h\sigma v_t(X_t) + \sqrt{h}\sigma z_t, \quad X_T \sim \mu \tag{5}$$

where the forward (the usual Itô's) and backward integrals indicate different endpoints at which we make the approximation. For any process $(Y_t)_t, (Z_t)_t$,

$$\int_0^T a_t(Y_t)\overrightarrow{dZ_t} \approx \sum_i a_{t_i}(Y_{t_i})(Z_{t_{i+1}} - Z_{t_i}), \quad \int_0^T a_t(Y_t)\overleftarrow{dZ_t} \approx \sum_i a_{t_{i+1}}(Y_{t_{i+1}})(Z_{t_{i+1}} - Z_{t_i}), \tag{6}$$

which in particular implies the martingale property

$$\mathbb{E}_{X \sim \overrightarrow{\mathbb{P}}^{\nu,f+\sigma u}}\left[\int_0^t a_s(X_s)\overrightarrow{dW_s}\right] = 0, \quad \mathbb{E}_{X \sim \overleftarrow{\mathbb{P}}^{\mu,f+\sigma v}}\left[\int_t^T a_s(X_s)\overleftarrow{dW_s}\right] = 0. \tag{7}$$

Note one can also view the backward process (3) as the time reversal of the following forward SDE with the additional score function: ($\hat{\rho}_t$ is the density of $\overleftarrow{\mathbb{P}}^{\mu,f+\sigma v}$ at time $t$)

$$d\hat{X}_t = (-\hat{f}_t(\hat{X}_t) - \sigma\hat{v}_t(\hat{X}_t) - \sigma^2\nabla\log\hat{\rho}_t(\hat{X}_t))dt + \sigma\overrightarrow{dW_t}, \quad \hat{X}_0 \sim \mu \tag{8}$$

with the hat denoting $t \to T - t$. Now time reversing while keeping the same path measure will give the following SDE:

$$dX_t = [f_t(X_t) + \sigma v_t(X_t)]dt + \sigma\overleftarrow{dW_t}, \quad X_T \sim \mu.$$

Therefore both (3) and (8) have the same path measure $\overleftarrow{\mathbb{P}}^{\mu,f+\sigma v}$. Forward/backward integral can be converted through

$$\int_0^T a_t(Y_t)\overrightarrow{dZ_t} = \int_0^T a_t(Y_t)\overleftarrow{dZ_t} - \sigma^2\int_0^T (\nabla \cdot a_t)(Y_t)\,dt, \tag{9}$$

which will be used repeatedly throughout.

*Remark* (Nelson's identity). The following relationship between time-reversal drifts for the SDE (2)-(3) is well known: $\overrightarrow{\mathbb{P}}^{\nu,f+\sigma u} = \overleftarrow{\mathbb{P}}^{\mu,f+\sigma v}$ iff $\overrightarrow{\mathbb{P}}_T^{\nu,f+\sigma u} = \mu$, and $\sigma v_t(X_t) = \sigma u_t(X_t) - \sigma^2\nabla\log(\overrightarrow{\mathbb{P}}_t^{\nu,f+\sigma u}) = \sigma u_t(X_t) - \sigma^2\nabla\log(\overleftarrow{\mathbb{P}}_t^{\mu,f+\sigma v})$ for all $t \in [0,T]$.

## 2.2 Related Approaches

We would like the two path measures (with the specified two end-point marginals) to agree progressing in either direction. The methods in Vargas et al. (2022); Zhang & Chen (2021) propose to set up a reference process with a similar structure as (1) with $f = 0$:

$$\nu(z) \underset{\mathbb{P}^{\eta,\sigma v}}{\overset{\mathbb{P}^{\nu,\sigma r}}{\rightleftharpoons}} \eta(x) \tag{10}$$

where $r$ is the drift for the reference process:

$$dX_t = \sigma r_t(X_t)dt + \sigma \overrightarrow{dW_t}, \ X_0 \sim \nu \Rightarrow (X_t)_{t \in [0,T]} \sim \overrightarrow{\mathbb{P}}^{\nu, \sigma r} \tag{11}$$

and $\eta = \overrightarrow{\mathbb{P}}_T^{\nu, \sigma r}$. The part worth attention is that we fix the backward process such that $\overleftarrow{\mathbb{P}}^{\mu, \sigma v}$ and $\overleftarrow{\mathbb{P}}^{\eta, \sigma v}$ differ only in initialization with the same ensuing dynamics driven by $v$. The method amounts to estimating loss of the following type:

$$\begin{aligned}
\mathcal{L}_{KL}(u) &= \mathbb{E}_{\overrightarrow{\mathbb{P}}^{\nu, \sigma u}} \left[ \log \left( \frac{d \overrightarrow{\mathbb{P}}^{\nu, \sigma u}}{d \overleftarrow{\mathbb{P}}^{\mu, \sigma v}} \right) \right] = \mathbb{E}_{\overrightarrow{\mathbb{P}}^{\nu, \sigma u}} \left[ \log \left( \frac{d \overrightarrow{\mathbb{P}}^{\nu, \sigma u}}{d \overrightarrow{\mathbb{P}}^{\nu, \sigma r}} \frac{d \overleftarrow{\mathbb{P}}^{\eta, \sigma v}}{d \overleftarrow{\mathbb{P}}^{\mu, \sigma v}} \right) \right] \\
&= \mathbb{E}_{\overrightarrow{\mathbb{P}}^{\nu, \sigma u}} \left[ \log \left( \frac{d \overrightarrow{\mathbb{P}}^{\nu, \sigma u}}{d \overrightarrow{\mathbb{P}}^{\nu, \sigma r}} \frac{d\eta}{d\mu} \right) + \log Z \right] \\
&= \mathbb{E}_{X \sim \overrightarrow{\mathbb{P}}^{\nu, \sigma u}} \left[ \int_0^T \frac{1}{2} \| u_s(X_s) - r_s(X_s) \|^2 ds + \log \left( \frac{d\eta}{d\mu} \right) (X_T) + \log Z \right].
\end{aligned} \tag{12}$$

This suggests initializing from $\nu$ to estimate the loss (12) with the current control $u^{\hat{\theta}}$ by simulating (2), followed by gradient descent to optimize the $\hat{\theta}$-parameterized control policy and iterating between the two steps can be a viable strategy, that could identify $u^{\theta^*}$ eventually and used to run (2) to draw samples from $\mu$. Note $\eta, \nu$ are simple distributions we have the freedom to pick, and the corresponding $r(\cdot)$ as well. The last step of (12) is a simple application of Girsanov's theorem for two SDEs with different drifts. This is highly suggestive of a control-theoretic interpretation Berner et al. (2022), where the cost (12) is composed of a running control cost and a terminal cost.

Common choice for $\nu$ include $\delta_0$ a fixed Dirac delta (this is technically speaking a half-bridge) or $\mathcal{N}(0, \sigma^2 I)$. The $r$-controlled reference process (11) can be either Brownian motion ($r_t = 0$) or an OU process ($r_t(x) = -\frac{x}{2\sigma}$). Both combinations will give $\eta$ as normally distributed. In the former case, the optimal drift $u^*$ with $f = 0$ is known to be the Föllmer drift Tzen & Raginsky (2019) and can be computed explicitly as (picking $T = 1$ for simplicity)

$$u_t^*(x) = \nabla \log \mathbb{E}_X \left[ \frac{d\mu}{d\eta}(X_1) | X_t = x \right] = \nabla \log \mathbb{E}_{z \sim \mathcal{N}(0, \sigma^2 I)} \left[ \frac{d\mu}{d\mathcal{N}(0, \sigma^2 I)}(x + \sqrt{1 - t} z) \right],$$

where the first expectation above is taken w.r.t the reference measure – Wiener process in this case; and it follows that $v_t^*(x) = x/t$ by Doob's $h$-transform. In the latter case, process (10) is simply in equilibrium. The purpose of introducing the reference process (10) is to cancel out terms that we don't have convenient estimate for when calculating the likelihood ratio. All proofs are deferred to Appendix A.

**Lemma 1** (Forward/Backward path-space likelihood ratio). *In the general case of reference*

$$dX_t = \sigma r_t^+(X_t)dt + \sigma \overrightarrow{dW_t}, \ X_0 \sim \Gamma_0 \qquad dX_t = \sigma r_t^-(X_t)dt + \sigma \overleftarrow{dW_t}, \ X_T \sim \Gamma_T \tag{13}$$

*where* $\overrightarrow{\mathbb{P}}^{\Gamma_0, \sigma r^+} = \overleftarrow{\mathbb{P}}^{\Gamma_T, \sigma r^-}$, *there are 4 terms in* $D_{KL}(\overrightarrow{\mathbb{P}}^{\nu, f + \sigma u} \| \overleftarrow{\mathbb{P}}^{\mu, f + \sigma v})$ *as*

$$\begin{aligned}
\mathbb{E}_{X \sim \overrightarrow{\mathbb{P}}^{\nu, f + \sigma u}} &\left[ \log(\frac{d\nu}{d\Gamma_0})(X_0) - \log(\frac{d\mu}{d\Gamma_T})(X_T) + \log Z \right. \\
&+ \frac{1}{2\sigma^2} \int_0^T (f_t + \sigma u_t - \sigma r_t^+)(X_t)^\top (f_t + \sigma u_t - \sigma r_t^+)(X_t) dt \\
&\left. - \frac{1}{\sigma^2} \int_0^T (f_t + \sigma v_t - \sigma r_t^-)(X_t)^\top (\frac{1}{2} f_t + \sigma u_t - \frac{\sigma}{2} v_t - \frac{\sigma}{2} r_t^-)(X_t) dt - \int_0^T \nabla \cdot (f_t + \sigma v_t - \sigma r_t^-)(X_t) dt \right].
\end{aligned}$$

*By picking* $\gamma^+, \gamma^- = 0$, *and Lebesgue base measure for* $\Gamma_0, \Gamma_T$, $D_{KL}(\overrightarrow{\mathbb{P}}^{\nu, f + \sigma u} \| \overleftarrow{\mathbb{P}}^{\mu, f + \sigma v})$ *becomes*

$$\mathbb{E}_{X \sim \overrightarrow{\mathbb{P}}^{\nu, f + \sigma u}} \left[ \int_0^T \frac{1}{2} \| u_t(X_t) - v_t(X_t) \|^2 - \nabla \cdot (f_t + \sigma v_t)(X_t) dt + \log \frac{d\nu(X_0)}{d\mu(X_T)} \right] + \log Z.$$

*The loss* (12) *enforce uniqueness (and correct marginals $\nu, \mu$) but doesn't impose optimality of the trajectory in any sense. The judicious choice of common $v$ and $\nu$ in* (1) *and* (10) *allow* (12) *to take on a control-theoretic interpretation.*

*Remark.* The approach in Chen et al. (2021a) which relies on training two drifts essentially amounts to solving $\min_{u,v} D_{KL}(\overrightarrow{\mathbb{P}}^{\nu,f+\sigma u}\|\overleftarrow{\mathbb{P}}^{\mu,f+\sigma v})$ jointly with (49)-(50), as shown in Richter et al. (2023), is non-unique.

In diffusion generative modeling applications, score-matching-based loss Hyvärinen & Dayan (2005) can be seen as minimizing the reverse KL over $s = (u - v)/\sigma$ by running the backward process (3) and using the Radon-Nikodym derivative in (46)-(47) with $f = 0, r^+ = r^- = 0$. It gives ($C$ is independent of $s$ below)

$$
\begin{aligned}
\mathcal{L}_{KL}(s) &= \mathbb{E}_{\overleftarrow{\mathbb{P}}^{\mu,\sigma v}} \left[ \log \left( \frac{d\overleftarrow{\mathbb{P}}^{\mu,\sigma v}}{d\overrightarrow{\mathbb{P}}^{\nu,\sigma v+\sigma^2 s}} \right) \right] \\
&= \mathbb{E}_{\overleftarrow{\mathbb{P}}^{\mu,\sigma v}} \left[ \int_0^T \frac{\sigma^2}{2} \|s_t(X_t)\|^2 \, dt + \sigma^2 \int_0^T \nabla \cdot s_t(X_t) \, dt \right] + C \\
&= \mathbb{E}_{\overleftarrow{\mathbb{P}}^{\mu,\sigma v}} \left[ \int_0^T \frac{\sigma^2}{2} \|s_t(X_t) - \nabla_x \log p_t^{\mu,\sigma v}(X_t|X_T = x_T)\|^2 \, dt \right] + C'
\end{aligned}
\tag{14}
$$

while fixing the backward drift $v$ for which $\nu$ is easy to sample from (e.g., an OU process). In the last transition above, we used the integration by parts identity

$$
\mathbb{E}_{\rho_t} \left[ \int_0^T s_t^\top \nabla \log \rho_t \, dt \right] = -\mathbb{E}_{\rho_t} \left[ \int_0^T \nabla \cdot s_t \, dt \right].
\tag{15}
$$

In practice there will be an ir-reducible loss since the terminals $\nu$ and $\overleftarrow{\mathbb{P}}_0^{\mu,\sigma v}$ don't match exactly for any finite $T$, but the dynamics $\overrightarrow{\mathbb{P}}^{\nu,\sigma v+\sigma^2 s}$ still makes sense as for two processes $(p_t)_t, (q_t)_t$ with different initialization, but same drift $\sigma v + \sigma^2 s^*$, $\partial_t D_{KL}(p_t\|q_t) = -\mathbb{E}_{p_t}[\|\nabla \log \frac{p_t}{q_t}\|^2] \leq 0$ contracts, although we have not been quantitative about the decay rate.

We therefore see that either the approach of (12) or (14) relies on fixing some aspect of the process involving (1) and (10) to restore uniqueness of the loss $\mathcal{L}$. In both cases, it results in an one-parameter loss, and the minimizer is not affected by the unknown constant $Z$ from the loss. Successfully optimizing $D_{KL}(\overrightarrow{\mathbb{P}}^{\nu,\sigma u}\|\overleftarrow{\mathbb{P}}^{\mu,\sigma v})$ or $D_{KL}(\overleftarrow{\mathbb{P}}^{\mu,\sigma v}\|\overrightarrow{\mathbb{P}}^{\nu,\sigma u})$ to zero error will imply that $u$ pushes $\nu$ to $\mu$ and $v$ vice versa, mimicking a noising/denoising reversible procedure.

## 3 Methodology

As we saw from Section 2, there are many degrees of freedom in transporting $\nu$ to $\mu$, which is not desirable for training purpose. This gives us the motivation to turn to losses that can enforce a canonical choice. We will adopt an "optimal control" perspective (which necessitates picking some reference) and leverage special properties of the SB problem (introduced below) to come up with valid control objectives for this effort.

### 3.1 Various Perspectives on Schrödinger Bridge

Most of these can be traced out in Léonard (2014); Chen et al. (2021b), which has its roots in statistical mechanics Schrödinger (1931) (and in modern terms, closely related to large-deviation results via Sanov's theorem). For us however, the following perspectives will be more fruitful. Consider over path space $C([0,T]; \mathbb{R}^d)$, given a base measure $Q$, the optimization problem

$$
P^* = \arg \min_{P_0=\nu, P_T=\mu} D_{KL}(P\|Q)
\tag{16}
$$

where we assume $Q$ admits SDE representation

$$
dX_t = f_t(X_t)dt + \sigma dW_t, \quad X_0 \sim \nu,
\tag{17}
$$

which is a slight generalization of the classical case where $f = 0$ is typically assumed.

(1) Mixture of pinned diffusions (weights given by $\pi^*$) can be seen by disintegration of path measure:

$$D_{KL}(P\|Q) = D_{KL}(P_{0T}\|Q_{0T}) + \int D_{KL}(P^{zx}\|Q^{zx})\, dP_{0T}(z,x)\,. \tag{18}$$

Since the only constraints are on the two end-points (16) is therefore equivalent to (19), by choosing $P^{zx} = Q^{zx}$. The optimal solution takes the form $P^* = \pi^* Q^{zx}$, which means one can sample from $(z,x) \sim \pi^* \in \Pi_{\nu,\mu}$, and sample from the bridges conditioned on the end-points at $t = 0, T$.

(2) It has the interpretation of entropy-regularized optimal transport Peyré & Cuturi (2019) when $f = 0$: in terms of static formulation because of (18), if the reference $Q$ is simply the Wiener process (accordingly $r(x|z) \propto e^{-\frac{1}{2T}\|x-z\|^2}$),

$$\begin{aligned}
\pi^*(z,x) &= \arg\min_{\pi_z=\nu,\pi_x=\mu} D_{KL}(\pi(z,x)\|r(z,x)) \tag{19}\\
&= \arg\min_{\pi_z=\nu,\pi_x=\mu} \mathbb{E}_{z\sim\nu}[D_{KL}(\pi(x|z)\|r(x|z))] + D_{KL}(\nu(z)\|r(z))\\
&= \arg\min_{\pi_z=\nu,\pi_x=\mu} \int \frac{1}{2}\|x-z\|^2 \pi(z,x)\, dxdz + T\int \pi(z,x)\log\pi(z,x)\, dxdz\,,
\end{aligned}$$

where the first term is nothing but the definition of the Wasserstein-2 distance (and optimal transport in the sense of Kantorovich). This is a Lagrangian description of the transport. In terms of dynamical formulation, via Girsanov's theorem on path measure, with the controlled dynamics as $dX_t = [f_t(X_t)+\sigma u_t(X_t)]dt+\sigma dW_t$, (16) can be reformulated as a *constrained* problem over $u$

$$P^* = \arg\min_P \mathbb{E}_{X\sim\overrightarrow{\mathbb{P}}^{\nu,f+\sigma u}}\left[\frac{1}{2}\int_0^T \|u_t(X_t)\|^2 dt \,\middle|\, \overrightarrow{\mathbb{P}}_T^{\nu,f+\sigma u} = \mu\right]\,. \tag{20}$$

Or a *regularized* Benamou-Brenier fluid-dynamics Benamou & Brenier (2000) analogy of the optimal transport

$$\inf_{\rho,v} \int_{\mathbb{R}^d}\int_0^T \left[\frac{1}{2}\|v_t(X_t)-\bar{v}_t(X_t)\|^2 + \frac{\sigma^4}{8}\left\|\nabla\log\frac{\rho_t(X_t)}{\bar{\rho}_t(X_t)}\right\|^2\right]\rho_t(X_t)\, dtdx \tag{21}$$

$$\text{s.t. } \frac{\partial\rho_t}{\partial t} + \nabla\cdot(\rho_t v_t) = 0, \rho_0 = \nu, \rho_1 = \mu \tag{22}$$

where above $\bar{v}_t = f_t - \frac{\sigma^2}{2}\nabla\log\bar{\rho}_t$ is the current velocity of the prior process, and we see the penalization results in an additional relative Fisher information term. Since $v^*$ is a velocity field in the continuity equation, it means a deterministic evolution (i.e., ODE) as

$$\dot{X}_t = v_t(X_t),\ X_0 \sim \nu$$

will have $X_t \sim \rho_t$, the optimal entropic interpolation flow, which gives an Eulerian viewpoint.

(3) Optimal control views the problem as steering $\nu$ at $t = 0$ to $\mu$ at $t = T$ with minimal control effort. The value function (i.e., optimal cost-to-go)

$$V(x,t) := \min_u \mathbb{E}_u\left[\frac{1}{2}\int_t^T \|u_s(X_s)\|^2 ds \,\middle|\, X_t^u = x, X_T^u \sim \mu\right] \tag{23}$$

with the expectation taken over the stochastic dynamics

$$dX_t^u = [f_t(X_t^u) + \sigma u_t(X_t^u)]dt + \sigma dW_t,\ X_0^u \sim \nu \tag{24}$$

should satisfy the Hamilton-Jacobi-Bellman equation via the dynamical programming principle

$$\frac{\partial V(x,t)}{\partial t} + f_t(x)^\top \nabla V(x,t) + \frac{\sigma^2}{2}\Delta V(x,t) - \frac{\sigma^2}{2}\|\nabla V(x,t)\|^2 = 0\,, \tag{25}$$

where the optimal control $u_t^*(X_t) = -\sigma \nabla V(X_t, t)$ and gives the unique solution $(\rho_t^u)_{t \geq 0}$ solving

$$\frac{\partial \rho_t^u}{\partial t} = -\nabla \cdot (\rho_t^u(f_t - \sigma^2 \nabla V_t)) + \frac{\sigma^2}{2} \Delta \rho_t^u, \quad \rho_0^u \sim \nu, \rho_T^u \sim \mu \,. \tag{26}$$

The correspondence between (23) and (25) can also be seen with the Feynman-Kac formula. Notice that

$$\min_u \ \mathbb{E}_u \left[ \int_0^T \frac{1}{2} \|u_t(X_t)\|^2 dt - \log \frac{\mu}{Q_T}(X_T) \right] = 0 \,,$$

which also gives an intuitive explanation of the optimally-controlled process. This holds since

$$D_{KL}(P\|Q) = \mathbb{E}_u \left[ \int_0^T \frac{1}{2} \|u_t(X_t)\|^2 dt \right] = \mathbb{E}_u \left[ \frac{1}{2} \int_0^T \|u_t(X_t)\|^2 dt - \log \frac{\mu}{Q_T}(X_T) \right] + D_{KL}(\mu\|Q_T) \,,$$

which gives the claim by data processing inequality. The two coupled PDEs, one run forward in time (Fokker-Planck (26)), one run backward in time (HJB (25)) are also the first-order optimality condition of

$$\inf_{\rho,u} \ \int_{\mathbb{R}^d} \int_0^T \frac{1}{2} \|u_t(X_t)\|^2 \rho_t(X_t) \, dt dx \tag{27}$$

$$\text{s.t. } \frac{\partial \rho_t}{\partial t} + \nabla \cdot (\rho_t(f_t + \sigma u_t)) = \frac{\sigma^2}{2} \Delta \rho_t, \ \rho_0 = \nu, \rho_1 = \mu \tag{28}$$

where again the optimal $u^* = -\sigma \nabla V$ is of gradient type. Above the Laplacian is responsible for the diffusion part, and (27)-(28) is related to (21)-(22) via a change of variable. One might try to design schemes by forming the Lagrangian for the above (27)-(28), and solve the resulting saddle-point problem, but this deviates somewhat from our pathwise narrative.

*Remark.* Compared to the Langevin SDE $dX_t = -\nabla f(X_t)dt + \sqrt{2}dW_t$, which only involves forward-evolving density characterization and reaches equilibrium as $T \to \infty$, the controlled SDE (24) is time-inhomogeneous and involves two PDEs (25)-(26). Langevin also has a backward Kolmogorov evolution for the expectation of a function $g$: let $V(x,t) = \mathbb{E}[g(X_T)|X_t = x]$ we have $\partial_t V(x,t) - \nabla f(x)^\top \nabla V(x,t) + \Delta V(x,t) = 0$ with $V(x,T) = g(x)$, and is de-coupled from the Fokker-Planck equation $\partial_t \rho_t - \nabla \cdot (\rho_t \nabla f) - \Delta \rho_t = 0$ with $\rho_0 = \nu$ in this case.

(4) The dual potentials $\phi, \psi$ yield optimal forward/backward drifts: It holds that the optimal curve admits representation

$$\log \rho_t = \log \phi_t + \log \psi_t \quad \text{for all } t \tag{29}$$

and solving the boundary-coupled linear PDE system on the control

$$\frac{\partial \phi_t}{\partial t} = -\frac{\sigma^2}{2} \Delta \phi_t - \nabla \phi_t^\top f_t, \quad \frac{\partial \psi_t}{\partial t} = \frac{\sigma^2}{2} \Delta \psi_t - \nabla \cdot (\psi_t f_t) \quad \text{for } \phi_0 \psi_0 = p_{\text{prior}}, \phi_T \psi_T = p_{\text{target}} \tag{30}$$

gives two SDEs for the optimal curve in (16):

$$dX_t = [f_t(X_t) + \sigma^2 \nabla \log \phi_t(X_t)]dt + \sigma \overrightarrow{dW_t}, \ X_0 \sim \nu \tag{31}$$

$$dX_t = [f_t(X_t) - \sigma^2 \nabla \log \psi_t(X_t)]dt + \sigma \overleftarrow{dW_t}, \ X_T \sim \mu \,. \tag{32}$$

Note that equations (31)-(32) are time reversals of each other and obey Nelson's identity thanks to (29). The transformation (29)-(30) that involves $(\rho_t^*, u_t^*) = (\rho_t^*, -\sigma \nabla V_t) \mapsto (\phi_t, \psi_t)$ is a typical $\log \leftrightarrow \exp$ Hopf-Cole change-of-variable from (25)-(26). These PDE optimality results can be found in Caluya & Halder (2021).

(5) Factorization of the optimal coupling: in fact it is always the case that

$$\frac{d\pi^*}{dr}(X_0, X_T) = e^{f(X_0)} e^{g(X_T)} \quad r\text{-}a.s. \,. \tag{33}$$

Moreover under mild conditions if there exists $\pi, f, g$ for which such decomposition holds and $\pi_0 = \nu, \pi_T = \mu$, $\pi$ must be optimal – such condition (33) is *necessary and sufficient* for characterizing the solution to the SB problem. (33) together with (18) give that (which can also be thought of as re-weighting on the path space)

$$\frac{d\pi^*}{dr}(X_{0:T}) = e^{f(X_0)}e^{g(X_T)} \quad r\text{-a.s.} \,, \tag{34}$$

obeying the Schrödinger system

$$\int e^{f(X_0)}r(X_{0T})e^{g(X_T)}dX_T = p_{\text{prior}}(X_0)\,, \quad \int e^{f(X_0)}r(X_{0T})e^{g(X_T)}dX_0 = p_{\text{target}}(X_T)\,. \tag{35}$$

Using Doob's $h$-transform, one can write the transition kernel of $\pi^*$ as a twisted dynamic of the prior $r$, and the $\phi, \psi$ in (31)-(32) can also be expressed as a conditional expectation:

$$\phi_t(x) = \int e^{g(X_T)}r(X_T|X_t = x)\,dX_T = \int_{\mathbb{R}^d} \phi_T(X_T)r(X_T|X_t = x)\,dX_T\,,$$

$$\psi_t(x) = \int e^{f(X_0)}r(X_t = x, X_0)\,dX_0 = \int_{\mathbb{R}^d} \psi_0(X_0)r(X_t = x|X_0)\,dX_0$$

for $t \in [0, T]$, where the consistency of

$$e^{f(X_0)}r(X_0) = \psi_0(X_0),\ e^{g(X_T)} = \phi_T(X_T)\ \Rightarrow\ \pi^*(X_0, X_T) = \psi_0(X_0)r(X_T|X_0)\phi_T(X_T)$$

can be verified by (35) and (29). But in general, the transition kernel of the un-controlled process $r(\cdot|\cdot)$ (or $\phi_T, \psi_0$) is not available analytically for implementation as such. But it suggests there's a backward-forward SDE dynamics one could write for $\phi_t, \psi_t$ with drift $f_t$ initialized at $\phi_T, \psi_0$ respectively, as hinted by (30) as well, if we view them as densities.

(6) Equation (31) means the optimal $v_t^*$ in the continuity equation (22) is $f_t + \sigma^2 \nabla \log \phi_t - \frac{\sigma^2}{2}\nabla \log \rho_t = f_t + \frac{\sigma^2}{2}\nabla \log \frac{\phi_t}{\psi_t}$, which gives the ODE probability flow for this dynamics (with $X_0 \sim \nu$)

$$dX_t = f_t(X_t) + \frac{\sigma^2}{2}(\nabla \log \phi_t(X_t) - \nabla \log \psi_t(X_t))\,dt\,, \tag{36}$$

in the sense that the time marginals $\rho_t^{(36)} = \rho_t^{(31)} = \rho_t^{(32)} = P_t^{(16)}$ all agree. Moreover, using the instantaneous change of variables formula Chen et al. (2018), we also have

$$\log \rho_T(X_T) = \log \rho_0(X_0) - \int_0^T \nabla \cdot f_t(X_t)\,dt - \frac{\sigma^2}{2}\int_0^T \nabla \cdot (\nabla \log \phi_t(X_t) - \nabla \log \psi_t(X_t))\,dt\,, \tag{37}$$

which will be useful in Proposition 2 for estimating the normalizing constant. Note that in (37) both the density and the point at which we are evaluating is changing.

*Remark.* The stochastic control formulation makes it clear that the trajectory we are trying to recover is a meaningful one in the sense of minimal effort. If one were to switch the order of $P$ and $Q$ in (16), the optimal control problem becomes (e.g., for $f = 0$)

$$\inf_u\ \mathbb{E}\left[\int_0^T \frac{1}{2}\|u_t(X_t)\|^2 dt\right]$$

$$\text{s.t. } dX_t = \sigma u_t(X_t)dt + \sigma dW_t, X_0 \sim \nu, X_T \sim \mu$$

for the expectation taken over the reference process, instead of the controlled state density $\rho_t^u$, which is not very intuitive. The slightly non-conventional aspect of this control problem is the fixed terminal constraint.

To briefly summarize, all these different viewpoints explore the deep connections between PDEs (controls) and SDEs (diffusions) in one way or another.

### 3.2 Natural Attempts

In this section we discuss several natural attempts with the intention of adapting the SB formalism for sampling from un-normalized densities.

(1) Iterative proportional fitting/Sinkhorn relies on factorized form (34) of the optimal solution $\pi^*(z,x) = P_{0T}^*$ for iterative projection as

$$P^{(1)} = \arg\min_{Q\in\mathcal{P}(\nu,\cdot)} D_{KL}(Q\|P^{(0)}) = \frac{P^{(0)}\nu}{P_0^{(0)}}, \quad P^{(0)} = \arg\min_{Q\in\mathcal{P}(\cdot,\mu)} D_{KL}(Q\|P^{(1)}) = \frac{P^{(1)}\mu}{P_T^{(1)}} \tag{38}$$

i.e., only the end point differ and one solves half-bridges using drifts learned from the last trajectory transition rollout, initializing from either samples from $\mu(x)$ or $\nu(z)$. But since we don't have samples from $\mu$, neither the score (i.e., the drift) nor the proceeding IPF updates/refinements can be implemented. In fact, the first iteration of the IPF proposal in De Bortoli et al. (2021); Vargas et al. (2021) precisely corresponds to the score-based diffusion proposal in Song et al. (2020). However, there is a connection between IPF and Path space EM, which means that coordinate descent on the objective $\min_{\phi,\psi} D_{KL}(\overrightarrow{\mathbb{P}}^{\nu,f+\sigma^2\nabla\phi}\|\overleftarrow{\mathbb{P}}^{\mu,f+\sigma^2\nabla\psi})$, when initializing at $\phi = 0$ (i.e., the Schrödinger prior), is a valid strategy for solving (16).

**Lemma 2** (Optimization of drifts of EM)**.** *The alternating scheme initializing with $\phi_0 = 0$ converge to $\pi^*(z,x)$:*

$$\psi_n = \arg\min_\psi D_{KL}(\overrightarrow{\mathbb{P}}^{\nu,f+\sigma^2\nabla\phi_{n-1}}\|\overleftarrow{\mathbb{P}}^{\mu,f+\sigma^2\nabla\psi}) \tag{39}$$

$$\phi_n = \arg\min_\phi D_{KL}(\overrightarrow{\mathbb{P}}^{\nu,f+\sigma^2\nabla\phi}\|\overleftarrow{\mathbb{P}}^{\mu,f+\sigma^2\nabla\psi_n}). \tag{40}$$

*Moreover, both updates are implementable assuming samples from $\nu$ is available, which resolves non-uniqueness in an algorithmic manner (fixing one direction of the drift at a time).*

However, Lemma 2 above also shows that the prior only enters in the first step, therefore as it proceeds, the prior influence tends to be ignored as error accumulates – this aspect is different from our proposal in Proposition 1 below.

(2) Some alternatives to solve SB do not require analytical expression for $\mu$: diffusion mixture matching Peluchetti (2023) tilts the product measure $\nu\otimes\mu$ towards optimality gradually by learning a slightly different term than the score; data-driven bridge Pavon et al. (2021) aims at setting up a fixed point recursion on the SB system (35) for finding the optimal $\phi^*,\psi^*$, but both rely on the availability of samples from $\mu$ (as well as $\nu$) to estimate various quantities for implementation.

(3) Naively one may hope to reparametrize (31)-(32) as (with the time-reversal condition baked in)

$$\min_u D_{KL}(\overrightarrow{\mathbb{P}}^{\nu,\frac{\sigma^2}{2}\nabla\log\rho_t+\nabla u_t}\|\overleftarrow{\mathbb{P}}^{\mu,\nabla u_t-\frac{\sigma^2}{2}\nabla\log\rho_t})$$

to learn the drift $\nabla u$ (this is somewhat akin to annealed flow, and expected to have unique solution). This way, when the two path measures are optimized to agree, the resulting forward & backward drift necessarily obey Nelson's identity for the desired curve $\rho_t$. However, this only applies if we have access to the score $\nabla\log\rho_t$, for $(\rho_t)_{t\in[T]}$ the SB curve of density we wish to follow interpolating from $\nu$ to $\mu$. Learning both $\rho$ and $\nabla u$ will become similar to learning $\phi$ and $\psi$, as the approaches we study in Section 3.3. In a similar vein, if one chooses $\Gamma_0 = \nu, \Gamma_T = \mu$ and $r_t^+ = \sigma\nabla u_t^*, r_t^- = \sigma\nabla u_t^* - \sigma^2\nabla\log\rho_t$ the optimal SB process as reference, Lemma 1 asserts that the loss becomes

$$\arg\min_{\nabla\phi} D_{KL}(\overrightarrow{\mathbb{P}}^{\nu,\sigma\nabla\phi_t}\|\overleftarrow{\mathbb{P}}^{\mu,\sigma\nabla\phi_t-\sigma^2\nabla\log\rho_t}) = \arg\min_{\nabla\phi} \mathbb{E}\left[\int_0^T \|\nabla\phi_t - \nabla u_t^*\|^2 dt\right],$$

something we cannot estimate and optimize for without additional information.

(4) The work of Caluya & Halder (2021) investigated the case where the reference process has a gradient drift (i.e., $f = -\nabla U$) and reduce the optimal control task to solving a high-dimensional PDE subject to initial-value constraint (c.f. Eqn (33) and (47) therein). However, solving PDEs is largely regarded to be computationally more demanding than simulating SDEs.

### 3.3 Training Loss Proposal

Recall our focus is on solving the regularized optimal transport problem (16) between $\nu$ and $\mu$, by learning the controls on the basis of samples from $\nu$. Out of many joint couplings with correct marginals (i.e., transport maps), the choice of a reference path process will select a particular trajectory $\rho_t$ between $\nu$ and $\mu$.

The execution of this plan crucially hinges on two ingredients: (1) backward / forward change of measure formula established in Lemma 1; (2) properties of the optimal drifts for SB from Section 3.1, which in turn can be exploited for training the controls. Challenge, as emphasized before, is we need to be able to estimate the resulting loss and optimizing it guarantees convergence to the unique solution dictated by the SB.

Proposition 1 below serves as our main result. Both (1) and (4) are guided by "minimum-action" principle w.r.t a reference. (2) bases itself on a reformulation of the HJB PDE involving the optimal control, and (3) is grounded in the FBSDE system for SB optimality Chen et al. (2021a). In all cases, the objective is a two-parameter loss that also allows us to recover the score $\sigma^2 \nabla \log \rho_t$ from the returned solution. Note that variance is taken w.r.t the uncontrolled process in (2) and w.r.t the controlled process in (3). Losses (1) and (4), being conversion from the constrained problem (and each other), will need $\lambda$ to be picked relatively small so that the first part of the objective becomes 0 to identify the unique SB solution.

**Proposition 1** (Regularized Control Objective). *For the problem of* (16)*, the following losses are justified:*

*(1)* $\arg\min_{\nabla\phi,\nabla\psi} D_{KL}(\overrightarrow{\mathbb{P}}^{\nu,f+\sigma^2\nabla\phi} \| \overleftarrow{\mathbb{P}}^{\mu,f-\sigma^2\nabla\psi}) + \lambda \cdot \mathbb{E}_{X\sim\overrightarrow{\mathbb{P}}^{\nu,f+\sigma^2\nabla\phi}}\left[\int_0^T \frac{\sigma^2}{2}\|\nabla\phi_t(X_t)\|^2 dt\right]$

*(2)* $\arg\min_{\nabla\phi,\nabla\psi} D_{KL}(\overrightarrow{\mathbb{P}}^{\nu,f+\sigma^2\nabla\phi} \| \overleftarrow{\mathbb{P}}^{\mu,f-\sigma^2\nabla\psi}) +$

$$Var_{X\sim\overrightarrow{\mathbb{P}}^{\nu,f}}\left(\psi_T(X_T) - \psi_0(X_0) + \int_0^T (-\frac{\sigma^2}{2}\|\nabla\psi_t\|^2 + \nabla\cdot f_t - \sigma^2\Delta\psi_t)(X_t)\,dt - \sigma\int_0^T \nabla\psi_t(X_t)^\top dW_t\right)$$

$$or\ \ Var_{X\sim\overrightarrow{\mathbb{P}}^{\nu,f}}\left(\phi_T(X_T) - \phi_0(X_0) + \frac{\sigma^2}{2}\int_0^T \|\nabla\phi_t\|^2(X_t)\,dt - \sigma\int_0^T \nabla\phi_t(X_t)^\top dW_t\right)$$

*(3)* $\arg\min_{\phi,\psi}\ Var_{X\sim\overrightarrow{\mathbb{P}}^{\nu,f+\sigma^2\nabla\phi}}\left((\phi_T+\psi_T-\log\mu)(X_T)\right) + Var_{X\sim\overrightarrow{\mathbb{P}}^{\nu,f+\sigma^2\nabla\phi}}\left((\phi_0+\psi_0-\log\nu)(X_0)\right) +$

$$Var_{X\sim\overrightarrow{\mathbb{P}}^{\nu,f+\sigma^2\nabla\phi}}\left(\phi_T(X_T) - \phi_0(X_0) - \frac{\sigma^2}{2}\int_0^T \|\nabla\phi_t\|^2(X_t)\,dt - \sigma\int_0^T \nabla\phi_t(X_t)^\top dW_t\right) +$$

$$Var_{X\sim\overrightarrow{\mathbb{P}}^{\nu,f+\sigma^2\nabla\phi}}\left(\psi_T(X_T) - \psi_0(X_0) - \int_0^T \left(\frac{\sigma^2}{2}\|\nabla\psi_t\|^2 + \nabla\cdot(\sigma^2\nabla\psi_t - f_t) + \sigma^2\nabla\psi_t^\top\nabla\phi_t\right)(X_t)\,dt\right.$$

$$\left.- \sigma\int_0^T \nabla\psi_t(X_t)^\top dW_t\right)$$

*(4)* $\arg\min_{\nabla\phi_t,\nabla\log\rho_t} D_{KL}(\overrightarrow{\mathbb{P}}^{\nu,f+\sigma^2\nabla\phi_t} \| \overleftarrow{\mathbb{P}}^{\mu,f+\sigma^2\nabla\phi_t-\sigma^2\nabla\log\rho_t}) + \lambda \cdot \mathbb{E}_{\overrightarrow{\mathbb{P}}^{\nu,f+\sigma^2\nabla\phi}}\left[\int_0^T \frac{\sigma^2}{2}\|\nabla\phi_t(X_t)\|^2\,dt\right]$

*In the above,* $D_{KL}(\overrightarrow{\mathbb{P}}^{\nu,f+\sigma^2\nabla\phi} \| \overleftarrow{\mathbb{P}}^{\mu,f-\sigma^2\nabla\psi})$ *is*

$$\mathbb{E}_{X\sim\overrightarrow{\mathbb{P}}^{\nu,f+\sigma^2\nabla\phi}}\left[\int_0^T \frac{\sigma^2}{2}\|\nabla\phi_t(X_t) + \nabla\psi_t(X_t)\|^2 - \nabla\cdot(f_t - \sigma^2\nabla\psi_t)(X_t)dt + \log\frac{\nu(X_0)}{\mu(X_T)}\right] + C \qquad (41)$$

*and* $D_{KL}(\overrightarrow{\mathbb{P}}^{\nu,f+\sigma^2\nabla\phi_t} \| \overleftarrow{\mathbb{P}}^{\mu,f+\sigma^2\nabla\phi_t-\sigma^2\nabla\log\rho_t})$ *is*

$$\mathbb{E}_{\overrightarrow{\mathbb{P}}^{\nu,f+\sigma^2\nabla\phi}}\left[\log\frac{\nu(X_0)}{\mu(X_T)} + \int_0^T \frac{\sigma^2}{2}\|\nabla\log\rho_t(X_t)\|^2 + \sigma^2\nabla\cdot(\nabla\log\rho_t - \nabla\phi_t)(X_t) - \nabla\cdot f_t(X_t)\,dt\right] + C'. \quad (42)$$

*In all cases,* $X_0 = x_0 \sim \nu$ *is assumed given as initial condition. In particular, if the loss is* 0 *for (2) and (3), the resulting* $\nabla\phi^*, \nabla\psi^*$ *solve the Schrödinger Bridge problem from* $\nu$ *to* $\mu$*, that can in turn be used for sampling from* $\mu$*.*

It is tempting to draw a connection between the variance penalty in Proposition 1 and the log-variance divergence between two path measures w.r.t a reference

$$\mathrm{Var}_{\mathbb{P}^w}\left[\log\left(\frac{d\mathbb{P}^u}{d\mathbb{P}^v}\right)\right]$$

studied in Richter et al. (2023). The loss in (3) is different from log-variance divergence over path space

$$\mathrm{Var}_{\overrightarrow{\mathbb{P}}^{\nu,f+\sigma^2\nabla\phi}}\left[\log\left(\frac{d\overrightarrow{\mathbb{P}}^{\nu,f+\sigma^2\nabla\phi}}{d\overleftarrow{\mathbb{P}}^{\mu,f-\sigma^2\nabla\psi}}\right)\right],$$

which involve the sum of (53) and (54), and will not guarantee finding the optimal path, whereas the objective in (3) incorporates the dynamics of *two* controlled (and coupled) dynamics w.r.t $\overrightarrow{\mathbb{P}}^{\nu,f}$ that we know how to characterize optimality for from results in Section 3.1.

In fact, another way to view the variance regularizers in loss (2) and (3) is through the SDE representation of the controls from Lemma 3 below and observe that the variance condition precisely encodes the optimally-controlled dynamical information.

**Lemma 3** (SDE correspondence to SB optimality). *We have for the optimal forward drift $\nabla\phi_t$ and $X \sim \overrightarrow{\mathbb{P}}^{\nu,f}$ as in* (17),

$$d\phi_t(X_t) = -\frac{\sigma^2}{2}\|\nabla\phi_t(X_t)\|^2 dt + \sigma\nabla\phi_t(X_t)^\top dW_t,$$

*analogously for the optimal backward drift $-\nabla\psi_t$, along $X \sim \overrightarrow{\mathbb{P}}^{\nu,f}$,*

$$d\psi_t(X_t) = \left[\frac{\sigma^2}{2}\|\nabla\psi_t\|^2 - \nabla\cdot f_t + \sigma^2\Delta\psi_t\right](X_t)dt + \sigma\nabla\psi_t(X_t)^\top dW_t.$$

*Moreover, along the controlled forward dynamics $X \sim \overrightarrow{\mathbb{P}}^{\nu,f+\sigma^2\nabla\phi}$, the optimal control $\phi,\psi$ satisfy*

$$d\phi_t(X_t) = \frac{\sigma^2}{2}\|\nabla\phi_t(X_t)\|^2 dt + \sigma\nabla\phi_t(X_t)^\top dW_t,$$

$$d\psi_t(X_t) = \left[\frac{\sigma^2}{2}\|\nabla\psi_t\|^2 + \nabla\cdot(\sigma^2\nabla\psi_t - f_t) + \sigma^2\nabla\phi_t^\top\nabla\psi_t\right](X_t)dt + \sigma\nabla\psi_t(X_t)^\top dW_t.$$

*In the above, $\nabla\phi_t, \nabla\psi_t$ refer to the optimal forward/backward drift of*

$$dX_t = [f_t(X_t) + \sigma^2\nabla\phi_t(X_t)]dt + \sigma dW_t, X_0 \sim \nu,$$

$$dX_t = [f_t(X_t) - \sigma^2\nabla\psi_t(X_t)]dt + \sigma dW_t,\ X_T \sim \mu,$$

*and $\phi_T(X_T) + \psi_T(X_T) = \log p_{target}(X_T), \phi_0(X_0) + \psi_0(X_0) = \log\nu(X_0)$ for $X \sim \overrightarrow{\mathbb{P}}^{\nu,f+\sigma^2\nabla\phi}$.*

It is natural to ask if one could replace the variance regularizer $\mathrm{Var}(\cdot)$ with a moment regularizer $\mathbb{E}[|\cdot|^2]$ in e.g., loss (3). However, while the variance is oblivious to constant shift, the moment loss will require knowledge of the normalizing constant $Z$ of the target $\mu$ to make sense. We would also like to add that enforcing the drifts to take the gradient form is not strictly necessary for some parts of the losses.

*Remark.* The gradient for either $D_{KL}$ or Log-variance w.r.t. $u$ and $v$ (and the Monte-Carlo estimate thereof) can be found in e.g., Richter et al. (2023), where the authors show that log-variance has the additional benefit of having variance of gradient $= 0$ at the optimal $u^*, v^*$, which is not true for $D_{KL}$ in general, and has consequence for gradient-based updates such as Algorithm 1. Similar argument applies to the variance regularizer we consider (e.g., loss (2)). For this, we look at the Gâteaux derivative of the variance function $V$ in an arbitray direction $\tau$ since

$$\frac{\delta}{\delta u}V(u,v;\tau) := \frac{d}{d\epsilon}\bigg|_{\epsilon=0}V(u+\epsilon\tau,v) \Rightarrow \partial_{\theta_i}V(u_\theta,v_\gamma) = \frac{\delta}{\delta u}\bigg|_{u=u_\theta}V(u,v_\gamma;\partial_{\theta_i}u_\theta).$$

Now if $\hat{V}$ is the Monte-Carlo estimate of the variance of a random quantity $g$, it is always the case that (we use $\frac{\delta}{\delta u}(\cdot)_\tau$ to denote derivative in the $\tau$ direction)

$$\frac{\delta}{\delta u}\hat{V}(u,v;\tau) = \frac{\delta}{\delta u}(\hat{\mathbb{E}}[g(u,v)^2] - \hat{\mathbb{E}}[g(u,v)]^2)_\tau = 2\hat{\mathbb{E}}[g(u,v)\frac{\delta}{\delta u}g(u,v)_\tau] - 2\hat{\mathbb{E}}[g(u,v)]\frac{\delta}{\delta u}\hat{\mathbb{E}}[g(u,v)]_\tau \,,$$

therefore if $g(u^*, v^*) = 0$ almost surely for every i.i.d sample, such as the regularizer in loss (2), the derivative w.r.t the control $u$ in direction $\partial_{\theta_i} u_\theta$ is 0 at optimality, implying $\text{Var}(\partial_{\theta_i}\hat{V}(u_{\theta^*}, v_{\gamma^*})) = 0$. One can, of course, also replace $D_{KL}$ with log-variance for the first part of the loss.

### 3.4 Discretization and Practical Implementation

We include a word about practical implementation in this part. As in Proposition F.1 of Vargas & Nüsken (2023), it is possible to trade the divergence term for a backward integral and estimate $D_{KL}(\overrightarrow{\mathbb{P}}^{\nu,u}\|\overleftarrow{\mathbb{P}}^{\mu,v})$ (up to constant independent of $u,v$) as

$$\frac{1}{N}\sum_{i=1}^N\left[\log\frac{\nu(X_0^i)}{\mu(X_{K+1}^i)} + \sum_{k=0}^{K-1}\frac{1}{2\sigma^2(t_{k+1}-t_k)}\|X_k^i - X_{k+1}^i + v_{k+1}(X_{k+1}^i)(t_{k+1}-t_k)\|^2\right] \tag{43}$$

for

$$X_{k+1}^i = X_k^i + u_k(X_k^i)(t_{k+1}-t_k) + \sigma\sqrt{t_{k+1}-t_k}\cdot z_k^i, \; z_k^i \sim \mathcal{N}(0,I) \tag{44}$$

the Euler-Maruyama discretization of the forward process using $N$ Monte-Carlo samples, where $K = T/h$ if the stepsize $h = t_{k+1} - t_k$ for all $k$ is kept constant.

*Remark.* As observed in Vargas & Nüsken (2023), discretized version with backward integral (43)-(44) has the additional benefit of giving ELBO for the normalizing constant $Z$ of $p_{\text{target}}$ with the estimator

$$\hat{Z} = \frac{\mu(X_K)q^v(X_{0:K-1}|X_K)}{\nu(X_0)p^u(X_{1:K}|X_0)} \tag{45}$$

in the sense that $\mathbb{E}_{\nu(X_0)p^u(X_{1:K}|X_0)}[\log\hat{Z}] \leq \log(\mathbb{E}_{\nu(X_0)p^u(X_{1:K}|X_0)}[\hat{Z}]) = \log[\int \mu(X_K)dX_K] = \log(Z)$.

Putting everything together, we give the proposed algorithm below.

---

**Algorithm 1** Control Objective Training for Sampling

---

**Require:** Initial draw $(X_0^{i,(0)})_{i=1}^N \in \mathbb{R}^d \sim \nu$ independent, initial controls $u^{(0)}, v^{(0)}$
**Require:** Un-normalized density $\mu$, drift $f$, num of time steps $K$, num of iterations $T$
    **for** $t = 0, \cdots, T-1$ **do**
        Simulate (44) with current control $(u_k^{(t)})_{k=0,\cdots,K}$, obtain $(X_k^{n,(t)})_{k=0,\cdots,K}$ for $n = 1,\cdots,N$
        Estimate the loss (43)+discretized regularizer (c.f. Lemma 4 below) and the gradient w.r.t the two parameterized controls using the samples $(X_k^{n,(t)})_{k=0,\cdots K,n=1,\cdots,N}$
        Gradient update on the parameters to obtain $u^{(t+1)}$ and $v^{(t+1)}$
    **end for**
    **return** $X_K^{1,(T)}, \cdots, X_K^{N,(T)}$ as $N$ samples from $\mu$ with their importance weights $w^{u^{(T)}}(X_K^{n,(T)})$ (58)

---

In practice, with imperfect control from the training procedure, one can perform importance sampling to correct for the bias / improve on the estimate.

**Proposition 2** (Importance Sampling). *The following can be used to estimate the normalizing constant of the target density $p_{target}$:*

*(1) For the optimal $\phi^*, \psi^*$, with $X_t \sim \overrightarrow{\mathbb{P}}^{\nu,f+\sigma^2\nabla\phi^*}$,*

$$Z = \frac{\mu(X_T)}{\nu(X_0)}\exp\left(\frac{\sigma^2}{2}\int_0^T\nabla\cdot(\nabla\phi_t^* - \nabla\psi_t^*)(X_t)\,dt + \int_0^T\nabla\cdot f_t(X_t)dt\right).$$

*(2) For any potentially suboptimal $\phi, \psi$, with $X_t \sim \overrightarrow{\mathbb{P}}^{\nu, f + \sigma^2 \nabla \phi}$,*

$$Z = \mathbb{E}_{\overrightarrow{\mathbb{P}}^{\nu, f + \sigma^2 \nabla \phi}} \left[ \exp \left( -\frac{\sigma^2}{2} \int_0^T \|\nabla \phi_t + \nabla \psi_t\|^2 + \nabla \cdot (f_t - \sigma^2 \nabla \psi_t) dt - \sigma \int_0^T \nabla \phi_t + \nabla \psi_t dW_t - \log \frac{\nu(X_0)}{\mu(X_T)} \right) \right].$$

*With a sub-optimal control $\nabla \phi$ and $X_t \sim \overrightarrow{\mathbb{P}}^{\nu, f + \sigma^2 \nabla \phi}$, re-weighting can be used to get an unbiased estimator of a statistics $g$ as $\mathbb{E}_\phi[g(X_T) w^\phi(X_T)] = \mathbb{E}_{\phi^*}[g(X_T)] = \mathbb{E}_\mu[g]$ for*

$$w^\phi(X) = \exp \left( \log \frac{\mu}{\nu * \mathcal{N}(0, \sigma^2 T \cdot I)}(X_T) - \frac{1}{2\sigma^2} \int_0^T \|f_t + \sigma^2 \nabla \phi_t\|^2 dt - \frac{1}{\sigma} \int_0^T (f_t + \sigma^2 \nabla \phi_t) dW_t \right).$$

The following lemma provides a recipe for estimating the regularizer and the normalizing constant with discrete-time updates. We focus on loss (3) from Proposition 1, since most of the other parts are similar or straightforward to adapt.

**Lemma 4** (Discretized Loss and Estimator). *For $X \sim \overrightarrow{\mathbb{P}}^{\nu, f + \sigma^2 \nabla \phi}$,*

$$Var\left( \psi_T(X_T) - \psi_0(X_0) - \int_0^T \left( \frac{\sigma^2}{2} \|\nabla \psi_t\|^2 + \nabla \cdot (\sigma^2 \nabla \psi_t - f_t) + \sigma^2 \nabla \psi_t^\top \nabla \phi_t \right)(X_t) \, dt - \sigma \int_0^T \nabla \psi_t(X_t) \, dW_t \right)$$

*can be estimated as ($Var_N$ denotes empirical estimate of variance using $N$ samples)*

$$Var_N \left( \psi_K(X_{K+1}^i) - \psi_0(X_0^i) - \right.$$
$$\left. \sum_{k=0}^{K-1} \frac{1}{2\sigma^2 h} \|X_k^i - X_{k+1}^i + (f_{k+1} - \sigma^2 \nabla \psi_{k+1})(X_{k+1}^i) h\|^2 + \sum_{k=0}^{K-1} \frac{1}{2\sigma^2 h} \|X_{k+1}^i - X_k^i - f_k(X_k^i) h\|^2 \right),$$

*which compared to the KL estimator (43) has two terms. The importance-weighted $Z$-estimator from (57) can be approximated as*

$$\frac{1}{N} \sum_{i=1}^N \exp \left( \log \frac{\mu(X_K^i)}{\nu(X_0^i)} - \sum_{k=0}^{K-1} \frac{1}{2\sigma^2 h} \|X_k^i - X_{k+1}^i + (f_{k+1} - \sigma^2 \nabla \psi_{k+1})(X_{k+1}^i) h\|^2 + \frac{1}{2} \sum_{k=0}^{K-1} \|z_k^i\|^2 \right).$$

*In both cases for $i \in [N]$ independently,*

$$X_{k+1}^i = X_k^i + (f_k(X_k^i) + \sigma^2 \nabla \phi_k(X_k^i)) h + \sigma \sqrt{h} \cdot z_k^i, \ z_k^i \sim \mathcal{N}(0, I).$$

*Remark.* In Vargas & Nüsken (2023), the authors propose $\int_0^T \mathbb{E} \left| \partial_t \phi + f^\top \nabla \phi + \frac{\sigma^2}{2} \Delta \phi + \frac{\sigma^2}{2} \|\nabla \phi\|^2 \right| (X_t) dt$ as the HJB regularizer (c.f. (25)) on top of $D_{KL}(\overrightarrow{\mathbb{P}}^{\nu, f + \sigma^2 \nabla \phi} \| \overleftarrow{\mathbb{P}}^{\mu, f - \sigma^2 \nabla \psi})$ for loss (2) in Proposition 1. As is clear from the proof in Proposition 1, it is equivalent to the variance condition. There is, of course, a similar HJB for the backward drift $\nabla \psi$, which in this case will read $\partial_t \psi + f^\top \nabla \psi - \frac{\sigma^2}{2} \Delta \psi - \frac{\sigma^2}{2} \|\nabla \psi\|^2 + \nabla \cdot f = 0$. By trading a PDE constraint with a SDE one (based on likelihood ratio of path measures), we can avoid evaluating the divergence term for the discrete updates. It remains to note the additional benefit of vanishing variance of the gradient at the optimal control, as remarked earlier.

## 4 Conclusion

We detailed various connections between the seemingly disjoint areas of diffusion generative-modeling, optimal stochastic control and optimal transport, with the goal of sampling from high-dimensional, complex distributions in mind. This is orthogonal to MCMC-based approaches, and is accomplished by a more "learning-driven" methodology for the optimal control/drift under a suitable control objective. There are a lot of flexibility in the choice of the reference, optimality criteria, divergence metric etc. (that also makes the case for SB more compelling than existing diffusion-based samplers mentioned in Section 2.2), which we hope to report on in the future.

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

# A   Missing Proofs and Calculations

We give the omitted proof of Lemma 1 from Section 2 below, followed by Lemma 2 from Section 3.2.

*Proof of Lemma 1.* We give a few equivalent expressions for the Radon-Nikodym derivative, written in terms of the drifts. Starting with Proposition 2.2 of Vargas & Nüsken (2023), $\overrightarrow{\mathbb{P}}^{\nu,f+\sigma u}$ almost surely,

$$
\log\left(\frac{d\overrightarrow{\mathbb{P}}^{\nu,f+\sigma u}}{d\overleftarrow{\mathbb{P}}^{\mu,f+\sigma v}}\right)(X) = \log(\frac{d\nu}{d\Gamma_0})(X_0) - \log(\frac{d\mu}{d\Gamma_T})(X_T) + \log Z
$$

$$
+ \frac{1}{\sigma^2}\int_0^T (f_t + \sigma u_t - \sigma r_t^+)(X_t)\left(\overrightarrow{dX_t} - \frac{1}{2}(f_t + \sigma u_t + \sigma r_t^+)(X_t)dt\right) \tag{46}
$$

$$
- \frac{1}{\sigma^2}\int_0^T (f_t + \sigma v_t - \sigma r_t^-)(X_t)\left(\overleftarrow{dX_t} - \frac{1}{2}(f_t + \sigma v_t + \sigma r_t^-)(X_t)dt\right). \tag{47}
$$

One can use (9) to convert the backward integral to a forward one with an additional divergence term

$$
D_{KL}(\overrightarrow{\mathbb{P}}^{\nu,f+\sigma u}\|\overleftarrow{\mathbb{P}}^{\mu,f+\sigma v}) = \mathbb{E}_{X\sim\overrightarrow{\mathbb{P}}^{\nu,f+\sigma u}}\left[\log\left(\frac{d\overrightarrow{\mathbb{P}}^{\nu,f+\sigma u}}{d\overleftarrow{\mathbb{P}}^{\mu,f+\sigma v}}\right)(X)\right]
$$

$$
= \mathbb{E}_{X\sim\overrightarrow{\mathbb{P}}^{\nu,f+\sigma u}}\left[\log(\frac{d\nu}{d\Gamma_0})(X_0) - \log(\frac{d\mu}{d\Gamma_T})(X_T) + \log Z\right.
$$

$$
+ \frac{1}{2\sigma^2}\int_0^T (f_t + \sigma u_t - \sigma r_t^+)(X_t)^\top(f_t + \sigma u_t - \sigma r_t^+)(X_t)dt
$$

$$
\left. - \frac{1}{\sigma^2}\int_0^T (f_t + \sigma v_t - \sigma r_t^-)(X_t)^\top(\frac{1}{2}f_t + \sigma u_t - \frac{\sigma}{2}v_t - \frac{\sigma}{2}r_t^-)(X_t)dt - \int_0^T \nabla\cdot(f_t + \sigma v_t - \sigma r_t^-)(X_t)dt\right] \tag{48}
$$

$$
+ \mathbb{E}_{X\sim\overrightarrow{\mathbb{P}}^{\nu,f+\sigma u}}\left[\int_0^T (u_t - r_t^+)(X_t) - (v_t - r_t^-)(X_t)\overrightarrow{dW_t}\right]
$$

and the last term vanishes because of (7). Therefore we see that there are 2 boundary terms, and 3 extra terms corresponding to the forward/backward process. The loss of (12) clearly follows since only the second line and one term in the first line survive. By picking $\gamma^+, \gamma^- = 0$, and Lebesgue base measure for $\Gamma_0, \Gamma_T$, we get

$$
D_{KL}(\overrightarrow{\mathbb{P}}^{\nu,f+\sigma u}\|\overleftarrow{\mathbb{P}}^{\mu,f+\sigma v}) = \mathbb{E}_{X\sim\overrightarrow{\mathbb{P}}^{\nu,f+\sigma u}}\left[\int_0^T \frac{1}{2}\|u_t(X_t) - v_t(X_t)\|^2 - \nabla\cdot(f_t + \sigma v_t)(X_t)dt\right] \tag{49}
$$

$$
+ \mathbb{E}_{X\sim\overrightarrow{\mathbb{P}}^{\nu,f+\sigma u}}\left[\log\frac{d\nu(X_0)}{d\mu(X_T)} + \int_0^T (u_t - v_t)(X_t)\overrightarrow{dW_t}\right] + \log Z \tag{50}
$$

which agrees with Proposition 2.3 in Richter et al. (2023) up to a conventional sign in $v$, and is also the same as the ELBO loss in (Chen et al., 2021a, Theorem 4). Note the different expressions in (12) and (49)-(50) by picking distinct references (13) vs. (10).

One could also start with the forward SDE (8) and apply Girsanov's theorem with (2),(8) together with the chain rule for the KL to get

$$
D_{KL}(\overrightarrow{\mathbb{P}}^{\nu,f+\sigma u}\|\overleftarrow{\mathbb{P}}^{\mu,f+\sigma v}) = \mathbb{E}_{\overrightarrow{\mathbb{P}}^{\nu,f+\sigma u}}\left[\log\left(\frac{d\overrightarrow{\mathbb{P}}^{\nu,f+\sigma u}}{d\overleftarrow{\mathbb{P}}^{\mu,f+\sigma v}}\right)(X)\right]
$$

$$= \mathbb{E}_{\overrightarrow{\mathbb{P}}^{\nu,f+\sigma u}} \left[ \log \left( \frac{d\nu}{d\overleftarrow{\mathbb{P}}_0^{\mu,f+\sigma v}} \right)(X_0) + \frac{1}{2} \int_0^T \|u_t(X_t) - (v_t + \sigma\nabla\log\rho_t^{\mu,f+\sigma v})(X_t)\|^2 dt \right]. \tag{51}$$

To see this is equivalent to (49)-(50), we use Fokker-Planck on the process (8) to reach

$$\mathbb{E}_{\overrightarrow{\mathbb{P}}^{\nu,f+\sigma u}} \left[ \log \left( \frac{d\mu}{d\overleftarrow{\mathbb{P}}_0^{\mu,f+\sigma v}} \right)(X) - \log Z \right] = \mathbb{E}_{\overrightarrow{\mathbb{P}}^{\nu,f+\sigma u}} \left[ \log \left( \frac{d\overleftarrow{\mathbb{P}}_T^{\mu,f+\sigma v}}{d\overleftarrow{\mathbb{P}}_0^{\mu,f+\sigma v}} \right)(X) \right]$$

$$= \mathbb{E}_{\overrightarrow{\mathbb{P}}^{\nu,f+\sigma u}} \left[ \int_0^T -\nabla\cdot(f_t + \sigma v_t)(X_t) + \sigma(u_t - v_t)(X_t)^\top \nabla\log\rho_t^{\mu,f+\sigma v}(X_t) - \frac{\sigma^2}{2}\|\nabla\log\rho_t^{\mu,f+\sigma v}(X_t)\|^2 dt \right] \tag{52}$$

as by Itô's lemma for the process (2) with drift $f + \sigma u$,

$$\int_0^T \partial_t \log \overleftarrow{\mathbb{P}}_t^{\mu,f+\sigma v}(X_t^{f+\sigma u})\, dt = \int_0^T \frac{1}{\rho_t}\left( -\nabla\cdot(\rho_t(f+\sigma v)) + \frac{\sigma^2}{2}\Delta\rho_t + \nabla\rho_t^\top(f+\sigma u) \right) dt$$

$$+ \sigma\int_0^T \frac{\nabla\rho_t^\top}{\rho_t}dW_t + \int_0^T \frac{\sigma^2}{2}\Delta\log\rho_t\, dt$$

$$= \int_0^T -\frac{\nabla\rho_t^\top}{\rho_t}(f_t + \sigma v_t + \sigma^2\nabla\log\rho_t - f_t - \sigma u_t)$$

$$- \nabla\cdot(f_t + \sigma v_t + \sigma^2\nabla\log\rho_t) + \frac{\sigma^2}{2}\Delta\log\rho_t\, dt + \int_0^T \sigma\frac{\nabla\rho_t^\top}{\rho_t}\, dW_t,$$

which upon simple arranging, using (55) and taking expectation over $X^{f+\sigma u} \sim \overrightarrow{\mathbb{P}}^{\nu,f+\sigma u}$ give (52). Now adding up the previous two displays (51) and (52) using

$$\mathbb{E}_{\overrightarrow{\mathbb{P}}^{\nu,f+\sigma u}} \left[ \log\left( \frac{d\nu}{d\overleftarrow{\mathbb{P}}_0^{\mu,f+\sigma v}} \right) \right] = \mathbb{E}_{\overrightarrow{\mathbb{P}}^{\nu,f+\sigma u}} \left[ \log\left( \frac{d\nu}{d\mu} \right) + \log\left( \frac{d\mu}{d\overleftarrow{\mathbb{P}}_0^{\mu,f+\sigma v}} \right) \right]$$

finishes the proof. $\qquad\square$

*Proof of Lemma 2.* We have up to a constant independent of $\phi, \psi$, using Lemma 1, (39)-(40) give the loss

$$\psi_n \leftarrow \arg\min_\psi \int_0^T \mathbb{E}_{X\sim\overrightarrow{\mathbb{P}}^{\nu,f+\sigma^2\nabla\phi_{n-1}}} \left[ \frac{\sigma^2\|\nabla\psi\|^2}{2} - \sigma^2\nabla\phi_{n-1}^\top\nabla\psi - \nabla\cdot(\sigma^2\nabla\psi) \right](X_t)dt$$

$$\phi_n \leftarrow \arg\min_\phi \int_0^T \mathbb{E}_{X\sim\overrightarrow{\mathbb{P}}^{\nu,f+\sigma^2\nabla\phi}} \left[ \frac{\sigma^2}{2}\|\nabla\phi - \nabla\psi_n\|^2 - \nabla\cdot(f+\sigma^2\nabla\psi_n) \right](X_t)dt + \mathbb{E}\left[ \log\frac{\nu(X_0)}{\mu(X_T)} \right]$$

where updating $\phi$ bears resemblance to (12), and updating $\psi$ is akin to the score matching loss (14). The equivalence to (38) assuming $\phi, \psi$ is expressive enough (i.e., $(\phi^*, \psi^*) = (\phi_{\theta^*}, \psi_{\gamma^*})$) and $\phi_0 = 0$ is shown in Vargas & Nüsken (2023) and relies on the fact that $\phi, \psi$ update (39)-(40) solve respectively

$$\nabla\psi_n = \nabla\phi_{n-1} - \nabla\log\rho_t^{\nu,f+\sigma^2\nabla\phi_{n-1}}, \quad \nabla\phi_n = \nabla\psi_n - \nabla\log\rho_t^{\mu,f+\sigma^2\nabla\psi_n},$$

which fix one side of the marginal alternatively, while reversing the path-wise transition from the previous one. It is known (38) converges to $\pi^*$ as $n \to \infty$. $\qquad\square$

In Section 2.2 we also claimed that for two processes $(p_t)_t, (q_t)_t$ with different initialization, but same drift $\sigma v + \sigma^2 s$ where $s$ is the score function, $\partial_t D_{KL}(p_t\|q_t) = -\sigma^2/2 \cdot \mathbb{E}_{p_t}[\|\nabla\log\frac{p_t}{q_t}\|^2] \le 0$ contracts towards each other, when we run the generative process

$$dX_t = [\sigma v_t(X_t) + \sigma^2 s_t(X_t)]dt + \sigma\overrightarrow{dW_t}, \ X_0 \sim \nu \ne \overleftarrow{\mathbb{P}}_0^{\mu,\sigma v},$$

although the actual rate maybe slow. We give this short calculation here for completeness:

$$\partial_t D_{KL}(p_t\|q_t)$$

$$= \int (\partial_t p_t(x)) \log \frac{p_t(x)}{q_t(x)} \, dx + \int q_t(x) \left[ \frac{\partial_t p_t(x)}{q_t(x)} - \frac{p_t(x)\partial_t q_t(x)}{q_t^2(x)} \right] dx$$

$$= \int p_t \left[ \sigma v_t + \sigma^2 s_t - \frac{\sigma^2}{2} \nabla \log p_t(x) \right]^\top \nabla \log \frac{p_t(x)}{q_t(x)} dx - \int \frac{p_t(x)}{q_t(x)} \partial_t q_t(x) dx$$

$$= \int p_t \left[ \sigma v_t + \sigma^2 s_t - \frac{\sigma^2}{2} \nabla \log p_t(x) \right]^\top \nabla \log \frac{p_t(x)}{q_t(x)} dx - \int \nabla \frac{p_t(x)}{q_t(x)} \left[ \sigma v_t + \sigma^2 s_t - \frac{\sigma^2}{2} \nabla \log q_t(x) \right] q_t(x) dx$$

$$= \int \left[ \sigma v_t + \sigma^2 s_t - \frac{\sigma^2}{2} \nabla \log p_t(x) - \sigma v_t - \sigma^2 s_t + \frac{\sigma^2}{2} \nabla \log q_t(x) \right]^\top \nabla \log \frac{p_t(x)}{q_t(x)} \cdot p_t(x) dx$$

$$= -\frac{\sigma^2}{2} \int \left\| \nabla \log \frac{p_t(x)}{q_t(x)} \right\|^2 p_t(x) dx \leq 0 \,.$$

Immediately follows is the proof of our main results: Proposition 1 and Lemma 3 from Section 3.3.

*Proof of Proposition 1.* The first term (41)-(42) simply involves writing $D_{KL}$ in terms of the (current) controls using Lemma 1, up to constants. One can regularize to remove the constraints in (20) while still ensuring the right marginals via

$$\mathcal{L}(\nabla\phi, \nabla\psi) := D_{KL}(\overrightarrow{\mathbb{P}}^{\nu,\sigma^2\nabla\phi} \| \overleftarrow{\mathbb{P}}^{\mu,-\sigma^2\nabla\psi}) + \lambda R(\nabla\psi) \quad \text{or } D_{KL}(\overrightarrow{\mathbb{P}}^{\nu,\sigma^2\nabla\phi} \| \overleftarrow{\mathbb{P}}^{\mu,-\sigma^2\nabla\psi}) + \lambda' R'(\nabla\phi)$$

where $R(\cdot)$ can be either from reference measure as in (1) or from optimality condition on the control as in (2). In both cases, the first term is a $\rho$ constraint (time reversal consistency with correct marginals), and the second one a $\phi$-constraint (enforce optimality).

For (2), we give one direction of the argument for $\nabla\psi$ first. Let $\overrightarrow{\mathbb{P}}^{\nu,f}$ denote the path measure associated with $dX_t = f_t(X_t)dt + \sigma dW_t, X_0 \sim \nu$, then $\overrightarrow{\mathbb{P}}^{\nu,f}$ almost surely, the Radon-Nikodym derivative between $\overrightarrow{\mathbb{P}}^{\nu,f}$ and the controlled backward process is

$$\log \left( \frac{d\overrightarrow{\mathbb{P}}^{\nu,f}}{d\overleftarrow{\mathbb{P}}^{\mu,f-\sigma^2\nabla\psi_t}} \right)(X) = \log \frac{d\nu(X_0)}{d\mu(X_T)} + \int_0^T \frac{\sigma^2}{2}\|\nabla\psi_t\|^2 - \nabla \cdot (f - \sigma^2\nabla\psi_t)dt + \sigma \int_0^T \nabla\psi_t \, dW_t \,.$$

In the case when the variance (taken along the prior $X \sim \overrightarrow{\mathbb{P}}^{\nu,f}$)

$$\text{Var} \left( \psi_T(X_T) - \psi_0(X_0) - \int_0^T \frac{\sigma^2}{2}\|\nabla\psi_t\|^2 + \nabla \cdot f - \sigma^2\Delta\psi_t(X_t)dt - \int_0^T \sigma\nabla\psi_t \, dW_t \right) = 0 \,,$$

it implies that the random quantity is almost surely a constant independent of the realization, and

$$\log \left( \frac{d\overrightarrow{\mathbb{P}}^{\nu,f}}{d\overleftarrow{\mathbb{P}}^{\mu,f-\sigma^2\nabla\psi_t}} \right)(X) = \log \frac{d\nu(X_0)}{d\mu(X_T)} + \psi_T(X_T) - \psi_0(X_0) + \log Z \,,$$

from which using the factorization characterization (34), and the terminal constraint $\overleftarrow{\mathbb{P}}_0^{\mu,f-\sigma^2\nabla\psi_t} = \nu, \overleftarrow{\mathbb{P}}_T^{\mu,f-\sigma^2\nabla\psi_t} = \mu$ imposed by the first KL condition, concludes that $\overleftarrow{\mathbb{P}}^{\mu,f-\sigma^2\nabla\psi_t}$ must be the unique solution to the SB problem. The other direction on $\nabla\phi$ is largely similar, and one can show using Girsanov's theorem and the variance condition that along the prior $X \sim \overrightarrow{\mathbb{P}}^{\nu,f}$,

$$\log \left( \frac{d\overrightarrow{\mathbb{P}}^{\nu,f+\sigma^2\nabla\phi_t}}{d\overrightarrow{\mathbb{P}}^{\nu,f}} \right)(X) = \int_0^T -\frac{\sigma^2}{2}\|\nabla\phi_t\|^2 dt + \int_0^T \sigma\nabla\phi_t \, dW_t = \phi_T(X_T) - \phi_0(X_0) \,,$$

to conclude in the same way. Note that evaluating the variance regularizer only requires simulating from the base process (17).

In (3), for any $\phi_t$, the likelihood ratio along $dX_t = (f + \sigma^2 \nabla \phi_t)dt + \sigma dW_t, X_0 \sim \nu$ is

$$\log \left( \frac{d\overrightarrow{\mathbb{P}}^{\nu,f+\sigma^2 \nabla \phi_t}}{d\overrightarrow{\mathbb{P}}^{\nu,f}} \right)(X) = \int_0^T \frac{\sigma^2}{2} \|\nabla \phi_t\|^2 dt + \int_0^T \sigma \nabla \phi_t \, dW_t \,, \tag{53}$$

which according to (34), has to be equal to $\psi_0(X_0) + \phi_T(X_T) - \log \nu(X_0) = -\phi_0(X_0) + \phi_T(X_T) \; \mathbb{P}^{\nu,f} \; a.s. \Rightarrow \mathbb{P}^{\nu,f+\sigma^2 \nabla \phi} \; a.s.$ to be optimal, justifying

$$\mathrm{Var}_{X \sim \overrightarrow{\mathbb{P}}^{\nu,f+\sigma^2 \nabla \phi}} \left( \phi_T(X_T) - \phi_0(X_0) - \frac{\sigma^2}{2} \int_0^T \|\nabla \phi_t\|^2 (X_t) \, dt - \sigma \int_0^T \nabla \phi_t(X_t) \, dW_t \right).$$

In a similar spirit, for any $\psi_t$, along the same process $X \sim \overrightarrow{\mathbb{P}}^{\nu,f+\sigma^2 \nabla \phi_t}$,

$$\log \left( \frac{d\overrightarrow{\mathbb{P}}^{\nu,f}}{d\overleftarrow{\mathbb{P}}^{\mu,f-\sigma^2 \nabla \psi_t}} \right)(X) = \int_0^T \sigma^2 \nabla \psi_t^\top \nabla \phi_t + \frac{\sigma^2}{2} \|\nabla \psi_t\|^2 - \nabla \cdot (f_t - \sigma^2 \nabla \psi_t) dt + \int_0^T \sigma \nabla \psi_t dW_t$$
$$+ \log \left( \frac{d\nu(X_0)}{d\mu(X_T)} \right) + \log Z \,, \tag{54}$$

which again using (34), has to be equal to

$$-\psi_0(X_0) - \phi_T(X_T) + \log \nu(X_0) = -\psi_0(X_0) - \log p_{\mathrm{target}}(X_T) + \psi_T(X_T) + \log \nu(X_0)$$

to be optimal, yielding the anticipated variance regularizer. Additionally, in order to enforce the terminal constraint along $\overrightarrow{\mathbb{P}}^{\nu,f+\sigma^2 \nabla \phi_t}$, it suffices to sum up (53)-(54) to impose the time-reversal consistency, which means $\phi_T(X_T) + \psi_T(X_T) = \log p_{\mathrm{target}}(X_T)$ and $\phi_0(X_0) + \psi_0(X_0) = \log p_{\mathrm{prior}}(X_0) = \log \nu(X_0)$ necessarily.

In (4), we use $\nabla \phi_t$ and $\nabla \log \rho_t$ as optimization variables instead of the two drifts, and it follows from the dynamical formulation (27)-(28). One can also read (42) as

$$\partial_t \log \rho_t(X_t) = \frac{1}{\rho_t}(\partial_t \rho_t + \nabla \rho_t^\top \dot{X}_t)$$
$$= \frac{1}{\rho_t}[-\nabla \cdot (\rho_t(f_t + \sigma^2 \nabla \phi_t)) + \frac{\sigma^2}{2}\Delta \rho_t] + \frac{1}{\rho_t}\nabla \rho_t^\top (f_t + \sigma^2 \nabla \phi_t - \frac{\sigma^2}{2}\nabla \log \rho_t)$$
$$= -\sigma^2 \Delta \phi_t + \frac{\sigma^2}{2}\Delta \log \rho_t - \nabla \cdot f_t \,,$$

which upon integrating and using integration by parts (15) give the loss. This first part establishes a particular relationship between the two variables, and the second part enforces optimality among all curves transporting between $\nu$ and $\mu$ driven by such dynamics – one can also switch this part to (21) if $\bar{v}_t$ is known. $\qquad \square$

*Proof of Lemma 3.* Using Itô's lemma, we have along the reference SDE $dX_t = f_t(X_t)dt + \sigma dW_t, X_0 \sim \nu$,

$$d\phi_t = \left[ \frac{\partial \phi_t}{\partial t} + \nabla \phi_t^\top f_t + \frac{\sigma^2}{2}\Delta \phi_t \right] dt + \sigma \nabla \phi_t^\top dW_t \,.$$

Now deducing from (30), since the optimal $\phi_t$ solves the PDE for all $(t, x) \in [0, T] \times \mathbb{R}^d$

$$\partial_t \phi_t = -f_t^\top \nabla \phi_t - \frac{\sigma^2}{2}\Delta \phi_t - \frac{\sigma^2}{2}\|\nabla \phi_t\|^2 \,,$$

substituting the last display into the previous one gives the result. Analogously, along the same reference process with Itô's lemma,

$$d\psi_t = \left[ \frac{\partial \psi_t}{\partial t} + \nabla \psi_t^\top f_t + \frac{\sigma^2}{2}\Delta \psi_t \right] dt + \sigma \nabla \psi_t^\top dW_t$$

and using the fact (30) that the optimal $\psi_t$ solves for all $(t, x) \in [0, T] \times \mathbb{R}^d$

$$\partial_t \psi_t = -\nabla \psi_t^\top f_t - \nabla \cdot f_t + \frac{\sigma^2}{2} \left( \|\nabla \psi_t\|^2 + \Delta \psi_t \right)$$

and plugging into the previous display yields the claim. In both of the PDE derivations above, we used the fact that for any $g \colon \mathbb{R}^d \to \mathbb{R}$,

$$\frac{1}{g} \nabla^2 g = \nabla^2 \log g + \frac{1}{g^2} \nabla g \nabla g^\top \Rightarrow \frac{1}{g} \Delta g = \Delta \log g + \|\nabla \log g\|^2 \tag{55}$$

by taking trace on both sides.

The second part of the lemma statement, where $X_t$ evolves along the optimally controlled SDE, follows from (Chen et al., 2021a, Theorem 3) up to a change of variable. Notice the sign change and the absence of the cross term in the dynamics for $\phi_t, \psi_t$ when $X \sim \overrightarrow{\mathbb{P}}^{\nu, f}$ vs. $X \sim \overrightarrow{\mathbb{P}}^{\nu, f + \sigma^2 \nabla \phi}$. $\qquad\square$

Proposition 2 and Lemma 4 are stated in Section 3.4, whose proof we give below.

*Proof of Proposition 2.* The optimal $\nabla \phi_t, \nabla \psi_t$ allow us to calculate $\log Z$ for $p_{\text{target}}$ using (41) as either (can be used without expectation, or with expectation and ignore the last term)

$$\log Z = \mathbb{E}_{\overrightarrow{\mathbb{P}}^{\nu, f + \sigma^2 \nabla \phi}} \left[ -\frac{\sigma^2}{2} \int_0^T \|\nabla \phi_t(X_t) + \nabla \psi_t(X_t)\|^2 dt + \int_0^T \nabla \cdot (f_t(X_t) - \sigma^2 \nabla \psi_t(X_t)) dt - \log \frac{d\nu(X_0)}{d\mu(X_T)} \right]$$

$$+ \mathbb{E}_{\overrightarrow{\mathbb{P}}^{\nu, f + \sigma^2 \nabla \phi}} \left[ -\sigma \int_0^T (\nabla \phi_t + \nabla \psi_t)(X_t) \overrightarrow{dW_t} \right] =: \mathbb{E}[-S] \, ;$$

or using (37), with the optimal $\nabla \phi^*, \nabla \psi^*$, since $Z$ is independent of $X_T$,

$$-\log Z = \log \nu(X_0) - \frac{\sigma^2}{2} \int_0^T \nabla \cdot (\nabla \log \phi_t^* - \nabla \log \psi_t^*)(X_t) \, dt - \int_0^T \nabla \cdot f_t(X_t) dt - \log \mu(X_T) \tag{56}$$

in which case the estimator is exact with $X_t$ following (31), or equivalently, (36). Notice (56) bears resemblance to Jarzynski's identity, since the term that's been integrated is essentially $\partial_t \log \rho_t(X_t)$.

In general for imperfect control, since $D_{KL}(\overrightarrow{\mathbb{P}}^{\nu, f + \sigma^2 \nabla \phi} \| \overleftarrow{\mathbb{P}}^{\mu, f - \sigma^2 \nabla \psi}) > 0$, $\log Z$ will only be lower bounded by $\mathbb{E}[-S]$. Using Lemma 1 however,

$$1 = \mathbb{E}_{\overrightarrow{\mathbb{P}}^{\nu, f + \sigma^2 \nabla \phi}} \left[ \left( \frac{d\overrightarrow{\mathbb{P}}^{\nu, f + \sigma^2 \nabla \phi}}{d\overleftarrow{\mathbb{P}}^{\mu, f - \sigma^2 \nabla \psi}} \right)^{-1} \right]$$

$$= \mathbb{E}_{\overrightarrow{\mathbb{P}}^{\nu, f + \sigma^2 \nabla \phi}} \left[ \exp \left( -\frac{\sigma^2}{2} \int_0^T \|\nabla \phi_t + \nabla \psi_t\|^2 + \nabla \cdot (f_t - \sigma^2 \nabla \psi_t) dt - \sigma \int_0^T \nabla \phi_t + \nabla \psi_t dW_t - \log \frac{\nu}{\mu} \right) \frac{1}{Z} \right] \tag{57}$$

$$=: \mathbb{E}[\exp(-S')/Z] \, ,$$

giving $Z = \mathbb{E}[\exp(-S')]$ using any (potentially sub-optimal) control $\nabla \phi, \nabla \psi$.

For the importance sampling, we use path weights suggested by the terminal requirement that $X_T^{\phi^*} \sim \mu$, therefore for any $\phi$ and $X \sim \overrightarrow{\mathbb{P}}^{\nu, f + \sigma^2 \nabla \phi}$,

$$w^\phi(X) = \frac{d\mathbb{P}_{X^{\phi^*}}}{d\mathbb{P}_{X^\phi}}(X) = \frac{d\mathbb{P}_{X^{\phi^*}}}{d\mathbb{P}_{X^r}}(X) \frac{d\mathbb{P}_{X^r}}{d\mathbb{P}_{X^\phi}}(X)$$

$$= \frac{d\mu}{d\mathbb{P}_{X_T^r}}(X_T) \frac{d\mathbb{P}_{X^r}}{d\mathbb{P}_{X^\phi}}(X)$$

$$= \exp\left( \log \frac{d\mu}{d\mathbb{P}_{X_T^r}}(X_T) - \frac{1}{2\sigma^2}\int_0^T \|f_t + \sigma^2\nabla\phi_t\|^2 dt - \frac{1}{\sigma}\int_0^T (f_t + \sigma^2\nabla\phi_t)\,dW_t \right), \qquad (58)$$

where we assumed the reference $r$ is drift-free:

$$dX_t = \sigma dW_t, \quad X_0 \sim \nu$$

hence $X_T^r \sim \nu * \mathcal{N}(0, \sigma^2 T \cdot I) \stackrel{d}{=} y + \sigma\sqrt{T}z$ for $y \sim \nu$, and $\mathbb{P}_{X_T^r}$ is easy to evaluate. Indeed, this choice guarantees

$$w^\phi(X_T^\phi) = \frac{d\mu}{d\mathbb{P}_{X^\phi}}(X_T^\phi)$$

therefore $\mathbb{E}_\phi[g(X_T^\phi)w^\phi(X_T^\phi)] = \int g(x)\mu(x)dx$ for any function $g$ on the terminal variable $X_T$ generated with control $\phi$. $\qquad\square$

*Proof of Lemma 4.* Similar to (44), we obtain the forward trajectory

$$X_{k+1}^i = X_k^i + (f_k(X_k^i) + \sigma^2\nabla\phi_k(X_k^i))(t_{k+1} - t_k) + \sigma\sqrt{t_{k+1} - t_k} \cdot z_k^i, \ z_k^i \sim \mathcal{N}(0, I) \qquad (59)$$

with Euler-Maruyama for each of the $i \in [N]$ samples. Rewriting the term (54) using (9), (46)-(47) and proceeding with the approximation (6) give

$$\frac{1}{\sigma^2}\int_0^T f_t(X_t)\overrightarrow{dX_t} - \frac{1}{2\sigma^2}\int_0^T \|f_t(X_t)\|^2 dt - \frac{1}{\sigma^2}\int_0^T (f_t - \sigma^2\nabla\psi_t)(X_t)\overleftarrow{dX_t} + \frac{1}{2\sigma^2}\int_0^T \|(f_t - \sigma^2\nabla\psi_t)(X_t)\|^2 dt$$

$$\approx \frac{1}{\sigma^2}\sum_{k=0}^{K-1} f_k(X_k)^\top(X_{k+1} - X_k) - \frac{1}{2\sigma^2}\|f_k(X_k)\|^2(t_{k+1} - t_k)$$

$$- \frac{1}{\sigma^2}(f_{k+1}(X_{k+1}) - \sigma^2\nabla\psi_{k+1}(X_{k+1}))(X_{k+1} - X_k) + \frac{1}{2\sigma^2}\|f_{k+1}(X_{k+1}) - \sigma^2\nabla\psi_{k+1}(X_{k+1})\|^2(t_{k+1} - t_k)$$

$$= \sum_{k=0}^{K-1} \frac{1}{2\sigma^2(t_{k+1} - t_k)}\|X_k - X_{k+1} + (f_{k+1} - \sigma^2\nabla\psi_{k+1})(X_{k+1})(t_{k+1} - t_k)\|^2$$

$$- \sum_{k=0}^{K-1} \frac{1}{2\sigma^2(t_{k+1} - t_k)}\|X_{k+1} - X_k - f_k(X_k)(t_{k+1} - t_k)\|^2,$$

which is the estimator for the variance regularizer with $(X_k)$ following (59).

It is also possible to avoid the divergence (i.e., Hutchinson's trace estimator) in the estimator (57) when discretizing by leveraging similar ideas. Direct computation using (46)-(47) tell us $Z \approx$

$$\frac{1}{N}\sum_{i=1}^N \exp\left( \log\frac{\mu}{\nu} - \sum_{k=0}^{K-1}\frac{1}{2\sigma^2(t_{k+1} - t_k)}\|X_k^i - X_{k+1}^i + (f_{k+1} - \sigma^2\nabla\psi_{k+1})(X_{k+1}^i)(t_{k+1} - t_k)\|^2 + \frac{1}{2}\|z_k\|^2 \right)$$

for the same Euler-Maruyama trajectory (59) as the estimator for the normalizing constant, where we used that the additional term

$$\sum_{k=0}^{K-1}\frac{1}{2\sigma^2(t_{k+1} - t_k)}\|X_{k+1} - X_k - (f_k + \sigma^2\nabla\phi_k)(X_k)(t_{k+1} - t_k)\|^2 = \frac{1}{2}\sum_{k=0}^{K-1}\|z_k\|^2$$

from the update. This can be understood as a ratio of two discrete chains as in (45). $\qquad\square$

# B Numerics

In this section, we summarize the main contributions of the work (Proposition 1 and Lemma 4) and offer numerical evidence on their advantages compared to existing proposals for solving an optimal trajectory problem in a typical MCMC setup.

## B.1 Algorithm specification

Below we pick the reference process to be a Brownian motion with $f = 0$ and $\lambda$ is a parameter that we tune for best performance, although in principle any $\lambda > 0$ should work.

1. PINN-regularization (Vargas & Nüsken (2023)): for $i = 1, \cdots, n$, and $Z \sim \mathcal{N}(0, I)$ independently draw in parallel

$$x_{k+1}^i = x_k^i + \sigma^2 h \nabla \phi(x_k^i, kh) + \sigma \sqrt{h} Z_k^i, \, x_0^i \sim \nu \tag{60}$$

for $k = 0, \cdots, K$ with stepsize $h = c/(K+1)$ for some $c \geq 1$. Using these trajectories $\{x_k^i\}$, minimize over $\phi$ and $s$,

$$\frac{1}{n} \sum_{i=1}^n \left[ \log \frac{\nu(x_0^i)}{\mu(x_{K+1}^i)} + \sum_{k=0}^K \frac{1}{2\sigma^2 h} \| x_k^i - x_{k+1}^i + \sigma^2 h \left( \nabla \phi(x_{k+1}^i, (k+1)h) - \nabla s(x_{k+1}^i, (k+1)h) \right) \|^2 \right]$$

$$+ \lambda \cdot \frac{1}{n} \sum_{i=1}^n \left[ \sum_{k=0}^K \left| \partial_t \phi(x_k^i, kh) + \frac{\sigma^2}{2} \Delta \phi(x_k^i, kh) + \frac{\sigma^2}{2} \| \nabla \phi(x_k^i, kh) \|^2 \right| \cdot h \right]. \tag{61}$$

Above $\phi(x, t), s(x, t)$ are two neural networks that take $t \in \mathbb{R}$ and $x \in \mathbb{R}^d$ as inputs and maps to $\mathbb{R}$, $\nabla$ is the space derivative w.r.t $x \in \mathbb{R}^d$ (i.e., derivative w.r.t the first input), $\partial_t$ is the time derivative (i.e., derivative w.r.t the second input). With the new $\phi, s$ we repeat (60) and (61) several times, and compute statistics using the latest samples $\{x_{K+1}^i\}_{i=1}^n$.

Another possibility is to replace first term of (61) with

$$\frac{1}{K+1} \text{Var}_n \Big[ \log \frac{\nu(x_0^i)}{\mu(x_{K+1}^i)} + \sum_{k=0}^K \frac{1}{2\sigma^2 h} \| x_k^i - x_{k+1}^i + \sigma^2 h \left( \nabla \phi(x_{k+1}^i, (k+1)h) - \nabla s(x_{k+1}^i, (k+1)h) \right) \|^2$$

$$- \sum_{k=0}^K \frac{1}{2\sigma^2 h} \| x_{k+1}^i - x_k^i - \sigma^2 h \nabla \phi(x_k^i, kh) \|^2 \Big], \tag{62}$$

which will be a log-variance divergence between the two path measures, and is oblivious to unknown normalizing constant. Similarly for (64) below.

2. Variance-regularization (loss (2) from Proposition 1): Simulate trajectories (60) exactly as before, but additionally simulate $\{y_k^i\}_{k=0}^K$ as follows and cache them: (this only needs to be done once)

$$y_{k+1}^i = y_k^i + \sigma \sqrt{h} Z_k^i, \, y_0^i \sim \nu. \tag{63}$$

Minimize over $\phi, s$ the following discretized loss

$$\frac{1}{n} \sum_{i=1}^n \left[ \log \frac{\nu(x_0^i)}{\mu(x_{K+1}^i)} + \sum_{k=0}^K \frac{1}{2\sigma^2 h} \| x_k^i - x_{k+1}^i + \sigma^2 h \left( \nabla \phi(x_{k+1}^i, (k+1)h) - \nabla s(x_{k+1}^i, (k+1)h) \right) \|^2 \right]$$

$$+ \frac{\lambda}{K+1} \cdot \text{Var}_n \left[ \phi(y_{K+1}^i, (K+1)h) - \phi(y_0^i, 0) + \sum_{k=0}^K \frac{1}{2\sigma^2 h} \left( \| y_{k+1}^i - y_k^i - \sigma^2 \nabla \phi(y_k^i, kh) h \|^2 - \| y_{k+1}^i - y_k^i \|^2 \right) \right]. \tag{64}$$

Notice the first term is the same as (61). Again repeat (60) and (64) several times, tracking the loss (64). Above $\text{Var}_n$ denotes the empirical variance across the $n$ trajectories $\{y_k^i\}_{i=1}^n$ of the quantity

inside $[\cdot]$. The loss (64), as the proof of Proposition 1 shows, comes from the fact that along the prior $X \sim \overrightarrow{\mathbb{P}}^{\nu,f}$,

$$\log\left(\frac{d\overrightarrow{\mathbb{P}}^{\nu,f+\sigma^2\nabla\phi_t}}{d\overrightarrow{\mathbb{P}}^{\nu,f}}\right)(X) = \int_0^T -\frac{\sigma^2}{2}\|\nabla\phi_t\|^2 dt + \int_0^T \sigma\nabla\phi_t^\top \, dW_t = \phi_T(X_T) - \phi_0(X_0), \qquad (65)$$

and we discretized the Radon-Nikodym derivative similiar to how it was done in the KL divergence $D_{KL}(\overrightarrow{\mathbb{P}}^{\nu,\sigma^2\nabla\phi}\|\overleftarrow{\mathbb{P}}^{\mu,-\sigma^2\nabla\psi})$ (also see Lemma 4 for similar derivation).

Instead of the regularizer (64), another discretization of condition (65) can be a TD-like regularizer similar to Liu et al. (2022):

$$...+\lambda\cdot\frac{1}{n}\sum_{i=1}^n\sum_{k=0}^K h\cdot\left|\phi(y_{k+1}^i,(k+1)h) - \phi(y_k^i,kh) + \frac{\sigma^2 h}{2}\|\nabla\phi(y_k^i,kh)\|^2 - \sigma\sqrt{h}\nabla\phi(y_k^i,kh)^\top Z_k^i\right|. \tag{66}$$

The loss above can also be justified with Lemma 3. In all (61), (64) and (66), we parameterize the forward / backward drift as $\sigma^2\nabla\phi, \sigma^2\nabla\phi - \sigma^2\nabla s$. A regularization on the backward drift involving $s$ is also possible for (61), (64), and (66), but should have similar performance.

3. Separately-controlled loss (loss (3) from Proposition 1): Simulate (60) as before, with the $n$ trajectories $\{x_k^i\}$. Minimize over $\phi, \psi$ the following discretized loss (c.f. Lemma 4):

$$\mathrm{Var}_n\left[\psi(x_{K+1}^i,(K+1)h) + \phi(x_{K+1}^i,(K+1)h) - \log\mu(x_{K+1}^i)\right] + \mathrm{Var}_n\left[\psi(x_0^i,0) + \phi(x_0^i,0) - \log\nu(x_0^i)\right] +$$

$$\frac{\lambda}{K+1}\cdot\mathrm{Var}_n\left[\psi(x_{K+1}^i,(K+1)h) - \psi(x_0^i,0) + \sum_{k=0}^K \frac{1}{2\sigma^2 h}\left(\|x_{k+1}^i - x_k^i\|^2 - \|x_k^i - x_{k+1}^i - \sigma^2 h\nabla\psi(x_{k+1}^i,(k+1)h)\|^2\right)\right]$$

$$+\frac{\lambda}{K+1}\cdot\mathrm{Var}_n\left[\phi(x_0^i,0) - \phi(x_{K+1}^i,(K+1)h) + \sum_{k=0}^K \frac{1}{2\sigma^2 h}\left(\|x_{k+1}^i - x_k^i\|^2 - \|x_{k+1}^i - x_k^i - \sigma^2 h\nabla\phi(x_k^i,kh)\|^2\right)\right].$$
$$(67)$$

Notice that it has 4 variance terms, and we alternate between simulating (60) and updating $\phi, \psi$ from (67) several times. For this loss, we parameterize the forward / backward drift as $\sigma^2\nabla\phi, -\sigma^2\nabla\psi$. One can also consider the SDE-based discretization for the last 2 terms but will incur additional Laplacian and divergence terms as suggested by Lemma 3. In contrast to losses (64) and (66), in the presence of reference drift $f \neq 0$, (67) will have the drift $f$ appearing in the objective in the 3rd term involving $\psi$, therefore it can alleviate the potential problem brought by prior forgetting.

We emphasize that the discretized regularizer (64) wouldn't be available without the path-wise stochastic process perspective. The FBSDE view (Lemma 3) will naturally lend to a TD-like regularizer as in Liu et al. (2022), although the actual form there differs as they consider dynamics with mean-field interaction. Similar comment applies to (67), where we *separately* impose optimality condition on

$$\log\left(\frac{d\overrightarrow{\mathbb{P}}^{\nu,f+\sigma^2\nabla\phi_t}}{d\overrightarrow{\mathbb{P}}^{\nu,f}}\right)(X) \quad \text{and} \quad \log\left(\frac{d\overrightarrow{\mathbb{P}}^{\nu,f}}{d\overleftarrow{\mathbb{P}}^{\mu,f-\sigma^2\nabla\psi_t}}\right)(X),$$

for $X \sim \overrightarrow{\mathbb{P}}^{\nu,f+\sigma^2\nabla\phi_t}$ to take factorized forms, as opposed to looking at their sum

$$\log\left(\frac{d\overrightarrow{\mathbb{P}}^{\nu,f+\sigma^2\nabla\phi}}{d\overleftarrow{\mathbb{P}}^{\mu,f-\sigma^2\nabla\psi}}\right)$$

only, effectively erasing the "SB optimality enforcement" part. The two discretized losses (64) and (67) are our main contributions. Both have the variance reduction (compared to TD (66) with stochasticity in the objective), without evaluating expensive Laplacian terms (as the PDE-based PINN approach (61) would require). We also highlight that the algorithm does not use gradient information from the target $\nabla\log\mu$ as e.g., Langevin would.

## B.2 Importance weighting

For the very last sampling SDE, we simulate (60) with the latest $\nabla\phi$, but re-weight the $n$ samples $\{x_{K+1}^i\}_{i=1}^n$ each with individual weights

$$w(x_{K+1}^i) = \frac{\mu(x_{K+1}^i)}{\rho(x_{K+1}^i)} \exp\left(-\sum_{k=0}^K \frac{1}{2\sigma^2 h}\|x_{k+1}^i - x_k^i\|^2 + \sum_{k=0}^K \frac{1}{2\sigma^2 h}\|x_{k+1}^i - x_k^i - \sigma^2 h\nabla\phi(x_k^i, kh)\|^2\right)$$

before taking average, i.e.,

$$\hat{\mathbb{E}}_\mu[g] = \frac{\frac{1}{n}\sum_{i=1}^n g(x_{K+1}^i)w(x_{K+1}^i)}{\frac{1}{n}\sum_{i=1}^n w(x_{K+1}^i)} \tag{68}$$

for a summary statistics $g: \mathbb{R}^d \mapsto \mathbb{R}$ we are interested in. Above $\rho$ is the Gaussian pdf of $\mathcal{N}(b_1, \sigma^2 c \cdot I + b_2)$ if $\nu = \mathcal{N}(b_1, b_2)$ and $h = c/(K+1)$. This estimator follows from Proposition 2 and is used for post-processing with a potentially suboptimal control $\nabla\phi$.

## B.3 Benchmark and Metrics

Given the computing resource we have available, it was challenging to scale up to high-dimensional problems with large neural network size. As a proof of concept, we consider three targets with different priors $\nu$:

- 2D Shifted Gaussian: $\mathcal{N}(x; [4, 4], I)$ with prior $\nu(x) = \mathcal{N}(x; [1, 1], 4I)$

- 2D Gaussian mixture model with 4 modes: $\frac{1}{4}\sum_{i=1}^4 \mathcal{N}(x; \mu_i, I)$ where $\{\mu_i\}_{i=1}^4 = \{-2, 2\} \times \{-2, 2\}$ and prior $\nu(x) = N(x; [0, 0], 4I)$

- 2D Gaussian mixture model with 9 modes: $\frac{1}{9}\sum_{i=1}^9 \mathcal{N}(x; \mu_i, I)$ where $\{\mu_i\}_{i=1}^9 = \{-5, 0, 5\} \times \{-5, 0, 5\}$ and prior $\nu(x) = N(x; [0, 0], 3.5^2 I)$

For each of the benchmarks and 4 losses, we plot the marginals and report the following:

- Absolute error in mean and relative error in standard deviation compared to the ground truth using importance-weighted Monte-Carlo estimates (68)

- The log normalizing constant $\log Z$ estimator (c.f. Lemma 4):

$$\log\left(\frac{1}{n}\sum_{i=1}^n \frac{\mu(x_{K+1}^i)}{\nu(x_0^i)} \exp\left[\sum_{k=0}^K \frac{1}{2}\|Z_k^i\|^2 - \frac{1}{2\sigma^2 h}\|x_k^i - x_{k+1}^i + \sigma^2 h(\nabla\phi(x_{k+1}^i, (k+1)h) - \nabla s(x_{k+1}^i, (k+1)h))\|^2\right]\right)$$

for method 1 and 2, and

$$\log\left(\frac{1}{n}\sum_{i=1}^n \frac{\mu(x_{K+1}^i)}{\nu(x_0^i)} \exp\left[\sum_{k=0}^K \frac{1}{2}\|Z_k^i\|^2 - \sum_{k=0}^K \frac{1}{2\sigma^2 h}\|x_k^i - x_{k+1}^i - \sigma^2 h\nabla\psi(x_{k+1}^i, (k+1)h)\|^2\right]\right)$$

for method 3, both using the final trajectory $\{x_k^i\}$ from (60).

## B.4 Result and Discussion

For the experiments, we parameterize $\phi, s : \mathbb{R}^d \times [0, c] \to \mathbb{R}$ as feed-forward neural networks where $c$ is a hyperparameter. The $\phi$ neural network has 2 hidden layers with 40 neurons each and the $s/\psi$ network has 1 hidden layer with 10 neurons. All affine transforms are followed by the GELU activation. Adam optimizer was used to train the models with $\beta_1 = 0.9, \beta_2 = 0.999$, and weight decay 0.01, where batches of trajectories are used for several steps of gradient updates in each epoch, before regenerating the $n$ trajectories and estimating the objective for the next round of updates on the NN parameters. Across all experiments,

we initialize $\phi(x_0^i, 0) \approx \log \mu(x_0^i)$. For fair comparison, the training process is stopped when the loss stops noticeably decreasing.

In practice we find the log-variance divergence (62) to perform better than the KL divergence, therefore proceed with this choice for the losses (61), (64), and (66).

In Table 1 and Figure 1 below we show the simulation results, followed by Table 2 where the hyperparameters used for the experiments are listed.

For a comparison on the computation speed of the four regularizers: using the GMM-9 configuration listed in Table 2, it took 126 seconds to train with PINN and TD, 118 seconds to train with Separate Control, and 677 seconds to train with the Variance regularizer. Generating trajectories requires much less time than evaluating the loss and its gradient. Computing each loss requires processing all $K$ states across $n$ trajectories. Combined with the number of training epochs and the number of updates per batch, we estimate the processing time per trajectory state as approximately

$$\frac{\texttt{training\_time}}{\texttt{epochs} \times \texttt{trajectories} \times \texttt{updates\_per\_batch} \times K}.$$

The stated processing time is therefore roughly $1.68 \cdot 10^{-3}$s for PINN, $1.81 \cdot 10^{-6}$s for Variance, $1.57 \cdot 10^{-6}$s for Separate Control, and $1.68 \cdot 10^{-6}$s for TD. The latter three are comparable and much faster than PINN whose Laplacian computation adds an order of magnitude processing time.

For reproducibility, the anonymous Github repository can be found at the following link:

https://anonymous.4open.science/r/ctrlds-643B/README.md

Table 1: Absolute errors for the importance-weighted mean $\mu$, relative errors for the importance-weighted standard deviation $\sigma$, and log evidence estimates $\log Z$ for various target distributions and objectives. The ground truth $\log Z$ is 0 for both GMM targets.

| Loss | Quantity | Gaussian | GMM-4 | GMM-9 |
|------|----------|----------|-------|-------|
| PINN | $\mu$ | 0.124 | 0.706 | 0.474 |
| | $\sigma$ | 0.035 | 0.543 | 0.238 |
| | $\log Z$ | $-12.526$ | $-5.716$ | $-3.604$ |
| Variance | $\mu$ | 0.158 | 0.091 | 0.292 |
| | $\sigma$ | 0.074 | 0.027 | 0.037 |
| | $\log Z$ | $-9.348$ | $-3.005$ | $-2.292$ |
| Separate Control | $\mu$ | 0.203 | 0.049 | 0.061 |
| | $\sigma$ | 0.251 | 0.008 | 0.003 |
| | $\log Z$ | $-0.361$ | $-1.851$ | $-2.710$ |
| TD | $\mu$ | 0.185 | 0.084 | 0.436 |
| | $\sigma$ | 0.179 | 0.011 | 0.085 |
| | $\log Z$ | $-17.690$ | $-3.134$ | $-2.255$ |

We see that on multi-modal targets, our separately controlled loss (67) is generally much better compared to the PINN loss (61), while TD (66) and variance regularizer (64) can sometimes be comparable. In practice, we also observe that the separately controlled loss is less sensitive to tuning parameters, as can also be observed by the smoother training loss trajectory.

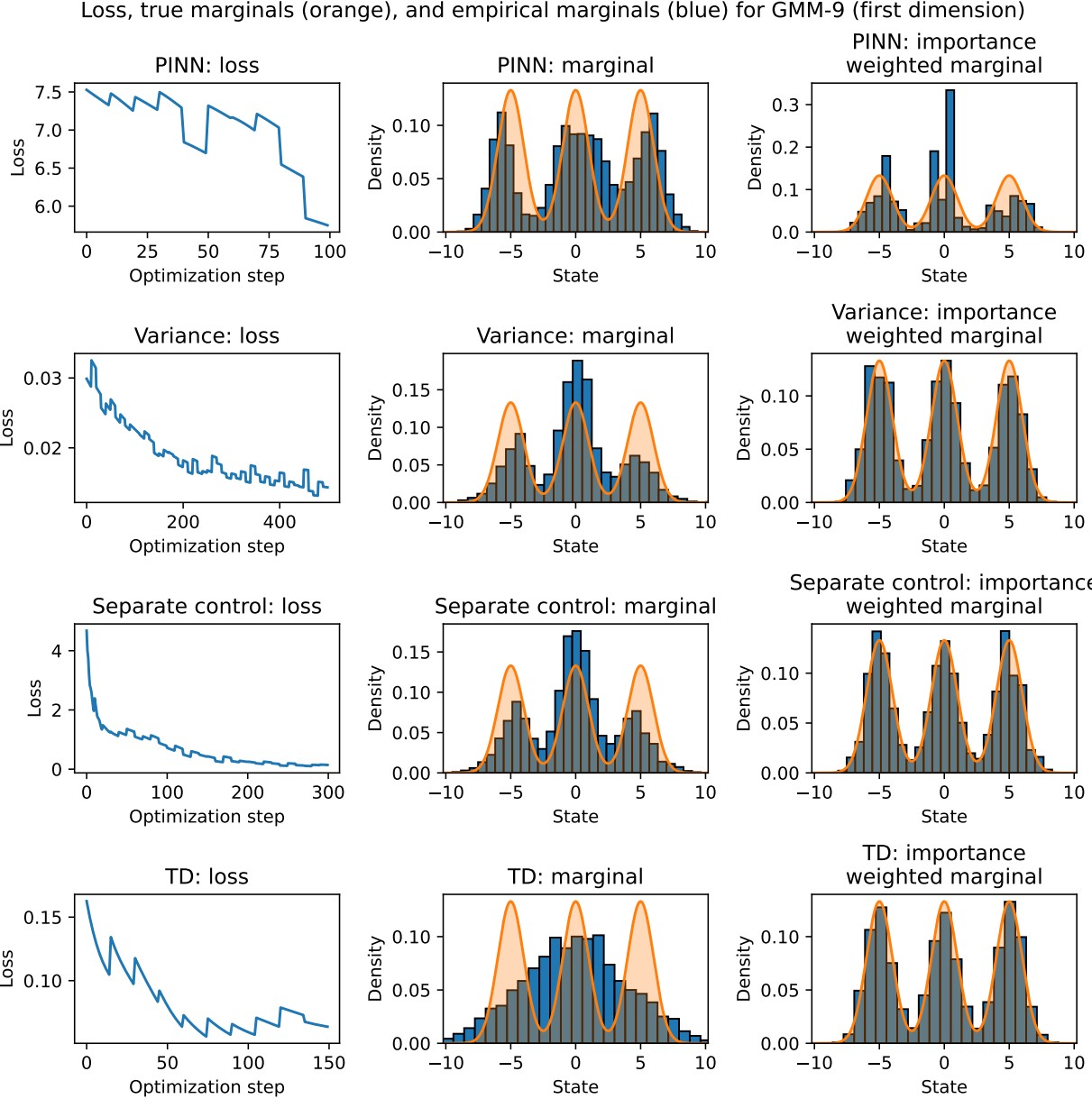

Figure 1: Gaussian mixture (9-mode): the second and third column show histograms of the samples for the first dimension with and without importance weights. Ground truth marginal density is shown in orange.

Table 2: Hyperparameter setting for different losses. Below $K$ is the trajectory length, $\lambda$ is the regularization coefficient, $\sigma$ is the diffusion coefficient in the SDE. Updates per batch refers to the number of passes we make over the $n$ trajectories in each epoch. Number of epoch is the number of times we re-generate the trajectories for training.

| Regularizer | Hyperparameter | Value (Gaussian) | Value (GMM-4 and GMM-9) |
|---|---|---|---|
| PINN | $K$ | 20 | 15 |
| | $\lambda$ | 1 | 1 |
| | c | 1.5 | 4 |
| | $\sigma$ | 1 | 1 |
| | learning rate | 0.005 | 0.0001 |
| | epochs | 20 | 10 |
| | trajectories (n) | 50 | 50 |
| | batch size | 50 | 50 |
| | updates per batch | 5 | 10 |
| Variance | $K$ | 150 | 150 |
| | $\lambda$ | 3 | 0.5 |
| | c | 2 | 2 |
| | $\sigma$ | 0.2 | 0.5 |
| | learning rate | 0.0001 | 0.0001 |
| | epochs | 50 | 50 |
| | trajectories (n) | 5000 | 5000 |
| | batch size | 500 | 5000 |
| | updates per batch | 10 | 10 |
| Separate Control | $K$ | 100 | 50 |
| | $\lambda$ | 1 | 3 (GMM-4) and 1 (GMM-9) |
| | c | 1.5 | 1 |
| | $\sigma$ | 1 | 1 |
| | learning rate | 0.005 | 0.005 |
| | epochs | 100 | 30 |
| | trajectories (n) | 500 | 5000 |
| | batch size | 500 | 5000 |
| | updates per batch | 10 | 10 |
| TD | $K$ | 100 | 100 |
| | $\lambda$ | 1 | 0.1 |
| | c | 1.5 | 1 |
| | $\sigma$ | 1 | 2 |
| | learning rate | 0.005 | 0.001 |
| | epochs | 100 | 10 |
| | trajectories (n) | 500 | 5000 |
| | batch size | 500 | 5000 |
| | updates per batch | 10 | 15 |

