# OpenReview forum: "Control, Transport and Sampling: The Benefit of a Reference Process"
_TMLR — Rejected by TMLR_

### Review · Reviewer_3jJF · 2023-11-01

**Summary Of Contributions:**

The study delves into a pivotal query, exploring the formulation of a bridge between two marginal distributions. Specifically, it employs instruments from the realm of stochastic differential equations to draw samples from an unnormalized density. Additionally, the work furnishes a thorough survey, encapsulating various methods, and forging links between executable algorithms, the Schrödinger bridge, and optimal control theory.

Nonetheless, the narrative proves challenging to navigate; many paragraphs are densely packed with information, rendering digestion tedious, and the transitions between sections lack fluidity, disrupting the reader's engagement and comprehension.

**Audience:**

Yes

**Claims And Evidence:**

Yes

**Requested Changes:**

See above

**Strengths And Weaknesses:**

1. Interestingly, many parts coincide with [1][2].
2. The introduction is short and does not cite any related work.
3. In eq-12, getting log Z in the second line from the first line is unclear to me.
4.  Following eq-12, the discussion of the optimal drift u^* in Tzen & Raginsky is strange. The authors have not set up optimality criteria and jump directly to the optimal solution.
5. In many equations, authors use new notations and it is easy to get readers confused eg.18.
6. Authors claim four algorithms are justified, but there are no justifications / explanations / reference.



[1] Lorenz Richter, Julius Berner, Guan-Horng Liu, Improved sampling via learned diffusions
[2] Anonymous. Improved sampling via learned diffusions. https://openreview.net/pdf?id=h4pNROsO06

---

> ### Author Response · Authors · 2023-11-03
>
> We would like to thank the reviewer for the feedback and below we address the comments:
>
> - Relationship to previous work: As the reviewer pointed out, the path-wise perspective from Section 2.2 that unify several existing generative-model-inspired algorithms was built upon those in [1-2], which we also explicitly cited in the paper. The contribution of the current work lies in the proposal of various new training losses in Proposition 1 for solving an optimal trajectory problem in a typical MCMC setup (i.e, without samples from the target). This part, as far as we understand, is not covered in the mentioned two works. If there are missing references that the reviewers believe we failed to cite or if we mis-interpreted the contribution of [1-2], we would be happy to learn -- we hope the relationships to several previous works are explained in the paper (in particular, Section 2.2 and 3.2) to a satisfying degree.
> - We apologize for the confusion in notation in eqn (12), $\mu$, throughout the paper, denotes the un-normalized target density, hence the extra $\log Z$ term coming from Radon-Nikodym derivative of two backward path measures, which only differ at the end-point. We will clarify the notation in the revision.
> - Discussion of the optimal $u^*$: Perhaps ``optimal" drift is the wrong word here. What we meant was by picking $r=0, \nu=\delta_0$, in which case $\eta$ is a Gaussian, the minimizer $u^*$ of equation (12) is given by the Follmer drift, which although perhaps still not clear at this stage, corresponds to the solution of a SB (i.e., stochastic optimal control) problem. Thanks for pointing this out.
> - Notation: In eqn (18), we are stating the chain rule formula for the KL divergence, where the first term corresponds to the terminal marginals, and in the second term $P^{zx}$ (resp. $Q^{zx}$) denotes the path measure $P$ (resp. $Q$) conditional on values at the two end points, and $P_{0T}(z,x)$ is the joint distribution at time $t=0,T$.
> - Justification: The fact that loss functions are valid training objectives are shown as part of the arguments in the proof of Proposition 1 (c.f. Page 16-17 of the current paper). More specifically, using the known optimality criteria on the SB controls, if one can train losses (2) and (3) to $0$, the drifts found necessarily have to be optimal. For losses (1) and (4), it will also depend on a judicious choice of the parameter $\lambda$ such that the first term imposing the path measure consistency can be minimized to $0$.
>
> We hope it clarifies some of the confusions. And please let us know if there are other parts that we could elaborate on  / needs clarification with. We appreciate the provided feedback.

---

### Review · Reviewer_ANoN · 2023-11-04

**Summary Of Contributions:**

This paper gives an exposition of some developments in diffusion-based sampling, along with what appear to be some new loss functions associated with the problem.

**Audience:**

Yes

**Broader Impact Concerns:**

None.

**Claims And Evidence:**

Yes

**Requested Changes:**

- Give numerical comparisons. Test the proposed methods and determine if any actually give a benefit over existing methods.

- Articulate the contributions more clearly and in the context of existing work. Currently, the introduction states "we propose control objectives that are amenable to tractable approximation (without access to data from the target, as classically considered in
the MCMC literature)" This is stated as a contribution, and it may be. But this work is not the first to make such a proposal, and so the comparative benefits of the proposed objectives should be discussed.

- Clean up the writing. There are lots of basic typos which should be fixed. Run a grammar checking tool.

**Strengths And Weaknesses:**

The biggest strength is that it gives a nice overview of the field and reviews interesting new developments and draws connections.

The biggest weakness is the relative lack of results. Aside from Proposition 1, most of the results appear to be fairly straightforward ways of rewriting existing results.

Beyond that, there is no comparison to existing methods. This might be okay if this were a brand new field. But it is not. Many of the cited papers, like (Vargas and Nuksen 2023) and (Zhang and Chen 2022), have solved variations on these problems and given numerical tests. So, finding new ways of thinking about the problem or defining new valid losses appears to have limited value without actually determining if any of these approaches give a benefit.

Even the surveyed connections to optimal transport and stochastic control are well-discussed in the cited literature. So, while the overview is a strength, many of these ideas are already explained the literature.

---

> ### Author Response · Authors · 2023-11-04
>
> We thank the reviewer for the feedback and here are a few things that we would like to clarify:
>
> - Review: We appreciate the comment. Just to be more clear on things: it isn't so much that we consider drawing connections of SB to optimal transport / stochastic control as new, as the reviewer mentioned, but (1) we bring them up to motivate why such an SB-based sampling approach is interesting to pursue, compared to other diffusion-inspired samplers that also take a path-wise perspective (c.f., those proposals in Section 2.2 that fix one side of the process for convenience); (2) the construction of our losses crucially rely on properties rendered by these perspectives.
> - Relationship to previous work: As the reviewer pointed out, the main contribution of the paper is Proposition 1, which, as far as we are aware, is not present in the literature, and is shown to have numerical advantages compared to existing algorithms/losses used to train for an SB-based sampler. Please refer to the Remark on page 12, and also the one at the end of Page 10 of the paper for comparison with Vargas & Nuksen 2023: It isn't that we consider a different SB formulation for performing sampling, but we believe it is a better loss to use for achieving the same goal. The setup considered in Zhang & Chen 2022 is a more restricted version of the SB problem compared to what we consider here. Please let us know if the reviewer believes there are references we missed to cite and/or if there are more comparisons that would be good to see. We hope the active research activities surrounding such approaches, as evidenced in the cited works of the paper, are also an indication of their interest.
> - Numerical comparison: Thank you for the suggestion. We will include numerics in the revised version in the coming days.

---

### Review · Reviewer_riKf · 2023-11-17

**Summary Of Contributions:**

Whilst the paper claims to establish connections between diffusion models and control, from what I can see all of these connections have already been highlighted in [1] and [2] so this reduces the verifiable contributions of this work to

1. Proposing variants of the BSDE regularizer discussed in [1]
2. Deriving the BSDE regularizer for the backward process
3. Providing several integration schemes for each of the above

The paper claims to "We aim to establish connections between diffusion-based sampling, optimal transport, and optimal (stochastic) control through their shared links to the Schrödinger bridge problem". However, the connections provided have already been discussed in [1,2,3,4].

[1] Vargas, F. and Nüsken, N., 2023, July. Transport, VI, and Diffusions. In ICML Workshop on New Frontiers in Learning, Control, and Dynamical Systems.

[2] Richter, L., Berner, J. and Liu, G.H., 2023. Improved sampling via learned diffusions. arXiv preprint arXiv:2307.01198.

[3] Chen, T., Liu, G.H. and Theodorou, E.A., 2021. Likelihood training of schr\" odinger bridge using forward-backward sdes theory. arXiv preprint arXiv:2110.11291.

[4] Liu, G.H., Chen, T., So, O. and Theodorou, E., 2022. Deep generalized Schrödinger bridge. Advances in Neural Information Processing Systems, 35, pp.9374-9388.

**Audience:**

Yes

**Broader Impact Concerns:**

None.

**Claims And Evidence:**

No

**Requested Changes:**

To make the contribution stronger it could be helpful to add some of the following:

1. Remove claims pertaining to connecting the forward-backwards KL framework to PIS or DDS as this has already been done in [1] and [2]
2. The proposed regularizers have already been discussed, the novelty of writing them down is not so huge on its own, so a proper ablation to showcase when these help in contrast to PINN, would be required.
3. The claim in the abstract "We aim to establish connections between diffusion-based sampling, optimal transport, and optimal (stochastic) control through their shared links to the Schrödinger bridge problem" as no novel connections pertaining to these items have been established in this work, existing connections have been discussed but that is not the same as establishing. If there is a new connection that I have missed I would be happy to discuss/revisit this point.

**Strengths And Weaknesses:**

# Strength

1. There is a clear computational advantage of using the proposed regularizers over PINN as one trades away the laplacian term in PINN. However, it remains unclear whether the reference forgetting aspects would continue to work as the Vargrad regularizers no longer explicitly include the prior drift (is absorbed when applying Itos lemma). So empirical evaluation is required or some sort of convincing error analysis.

# Weaknesses

1. The proposed vargrad SB regularizer for the forward process has already been discussed in [1]  (and a very closely related regularizer in  [4]) see remark 15 Equation 67 of https://openreview.net/pdf?id=Ay1b1W7Mjy
2. The paper states the **EM <=> IPF** result from [1] without reference and reads out as though it's a new result which it is not.
3. The connections to PIS (Path integral sampling/N-SFS) equations 10-12 are not new results and can be found both in [1] and [2]
4. Equations 14-15 providing further connections to score matching can also be found in the early Appendix of [1] , it is odd for this paper to redo these connections without reference. They are not new results
5. There are no compelling motivations presented for the new regularizer. Its nice to explore new ideas but without theoretical or empirical justification this contribution is incomplete.
6. A lot of the connections to control have been discussed in prior works [1,2] (e.g. see mean-field formulation in [1]) , whilst it is nice + pedagogical to have this (and can be worth including to aid a story), this on its own is not quite a contribution and has been discussed in the field for quite a bit now see [3].

---

> ### Author Response · Authors · 2023-11-20
>
> Thanks for the detailed comments - we appreciate the feedback. There are a few things perhaps we'd like to clarify.
>
> - Related work: Thanks for pointing them out and for the summary. The related works the reviewer mentioned are indeed cited in the current paper, and as we explicitly mentioned at the bottom of Page 2, the pathwise perspective from Section 2.2 that unifies several existing generative-model-inspired algorithms was built from those in [1] and [2], including equation (10),(12),(14-15) brought up by the reviewer (in our humble opinion, the reader would benefit from these expositions as context to appreciate the advantage of an SB-based approach and we weren't sure if it would be a good idea to avoid such discussions - please let us know if the reviewer has a better suggestion!). The main contribution of the current work lies in the proposal of various training losses in Proposition 1 for solving an optimal trajectory problem in a typical MCMC setup, along with their discretizations. We apologize for making the contribution not as clear as it could be. Moving forward, we will add a contribution section to make it easier to see, beyond just the sentence "Proposition 1 below serves as our main result" in the paragraph right before it as it stands now. As the reviewer also correctly pointed out, the connection to optimal control and optimal transport is known in the literature (we cited 2 review articles on the SB problem covering its relationship to other fields at the beginning of Section 3.1), but (1) we detail them to motivate why such an SB-based sampling approach is interesting to pursue, compared to other diffusion-inspired samplers that also take a path-wise perspective as those discussed in Section 2.2, which has both *uniqueness* and *optimality* baked in; (2) the construction of our losses / normalizing constant estimates crucially rely on properties rendered by these perspectives. Just to expand on things a bit, while [3] also draws a connection of SB to control, the proposed training loss itself, does not necessarily guarantee uniqueness.
> - "EM <-> IPF": Thanks for the comment - and yes we mention in the proof of Lemma 2 that the relationship was shown in [1] on Page 15 of the current paper. In hindsight, perhaps the reason for such confusion was due to the fact that some of the relationship to previous work was mentioned in the proof rather than the main text of the paper, therefore causing confusion at first reading. We apologize and will re-organize so these can be made clearer. Related to the reviewer's other comment, another advantage of a joint minimization procedure compared to an iterative IPF method is that while the equivalence of EM <-> IPF will be gone for neural network models not expressive enough (therefore the convergence does not immediately carry over), one could still estimate the normalizing constant / sample from the target with a sub-optimal control learned from the losses proposed in Proposition 1 with e.g., important weights.
> - Contribution / Motivation: In Proposition 1, losses (1), (2), (4) are based on regularizations on a KL-divergence imposing path-wise consistency with prescribed terminals. The additional regularization terms are responsible for identifying the optimal path. These are also the approaches taken in [1-4], where in all cases the quantity of interest is $\log\left(\frac{d\overrightarrow{\mathbb{P}}^{\nu,f+\sigma^2\nabla \phi}}{d\overleftarrow{\mathbb{P}}^{\mu,f-\sigma^2\nabla \psi}}\right)=\log \left(\frac{d\overrightarrow{\mathbb{P}}^{\nu,f+\sigma^2\nabla \phi_t}}{d\overrightarrow{\mathbb{P}}^{\nu,f}}\right)+\log\left(\frac{d\overrightarrow{\mathbb{P}}^{\nu,f}}{d\overleftarrow{\mathbb{P}}^{\mu,f-\sigma^2\nabla \psi_t}}\right)$, effectively erasing the "optimality enforcement" part. Loss (3), however, operates under a different principle, where we impose the forward / backward optimally-controlled processes w.r.t the reference separately, with the two terminal conditions explicitly incorporated. The fact that the loss is based on variance terms is particularly beneficial numerically with Monte-Carlo gradient estimates, since the variance of the gradient estimates is 0 at OPT. The corresponding discretization for the loss (3) is detailed in Lemma 4. Thanks for bringing Remark 15 of [1] on the forward process to our attention, we missed this discussion in the paper and will include this explicitly in the revision.
> - Numerical experiment comparing with PINN: Thanks for the suggestion! We will follow up with an update on the draft in the coming days.
> We appreciate any additional comments.

---

### Author Response · Authors · 2023-12-01
**Response to Reviewers**

Dear Reviewers:

We have uploaded a revision of the draft incorporating the suggestions on numerical experiments, while highlighting the contribution of the current work compared to several existing proposals that were brought up. To summarize, Proposition 1 and Lemma 4 are the main results of the paper (in particular the discretized losses (64) and (67)), and these are further detailed in the newly-added Appendix B of the current paper. In order to more easily see the changes that were made, we have avoided editing the main text and Appendix A of the paper beyond correcting minor typos. We will, however, implemented the various suggested changes and incorporated helpful comments from the reviewers in the final revision.

We again appreciate the comments and welcome any additional feedback.

Best,

Authors

---

### Decision · Action_Editor_JRpn · 2024-01-04

**Recommendation:** Reject

**Comment:**

While we acknowledge that the authors have made significant changes to their manuscript, it is still not ready for publication and requires more work. In particular, as pointed out by 2 out of 3 reviewers, it is not clear what is the scope of the paper. A lot of the writing addresses the background on Schrodinger Bridges, Optimal Control, Sampling and Diffusion Models which already exists in the literature (as Reviewer riKf points out, a lot of this background is covered notably in [1], including Equations (14)-(15) and a thorough discussion on the non-uniqueness in the absence of regularizers). The manuscript should make clearer the differences with the contributions of [1].

The core contribution of the work appears in Proposition 1. The authors should decide between a) a review paper of existing techniques (and in that case more experiments and discussions are required) b) a technical paper based on the proposed losses of Proposition 1 (and following), in that case more experiments are needed. In that sense, as pointed out by Reviewer riKf, Figure 1 is a step in the right direction. We encourage the authors to pursue this study by studying distributions for which benchmarks already exist (such as the Funnel distribution or the double well). This experimental work should be pursued and included in the main document. Exploring the mode collapse of the proposed methods (even in simple GNN settings such as the ones presented in [2]) would also constitute a great avenue for investigation.

[1] Vargas, F. and Nüsken, N., 2023. Transport, Variational Inference and Diffusions: with Applications to Annealed Flows and Schr" odinger Bridges. arXiv preprint arXiv:2307.01050.

[2] Midgley, L.I., Stimper, V., Simm, G.N., Schölkopf, B. and Hernández-Lobato, J.M., 2022. Flow annealed importance sampling bootstrap. arXiv preprint arXiv:2208.01893.

**Audience:**

Yes

**Claims And Evidence:**

No

**Resubmission Of Major Revision:**

The authors may consider submitting a major revision at a later time.